# Clonal evolutionary analysis reveals patterns of malignant transformation of Intraductal Papillary Mucinous Neoplasms of the pancreas

Antonio Pea [1,2,3,12] ✉, Xiaotong He [4,12], Rosie Upstill-Goddard[3], Claudio Luchini [5], Leonor Patricia Schubert Santana[3], Stephan Dreyer[3,6], Fraser Duthie[7], Nigel B. Jamieson [3,6], Colin J. McKay[6], Euan J. Dickson[6], Alessandra Pulvirenti[8], Selma Rebus[3], Paola Piccoli[5], Nicola Sperandio[9], Rita T. Lawlor [2,9], Michele Milella[2,10], Fieke E. M. Froeling [3], Roberto Salvia[1,2], Aldo Scarpa [4,9], Andrew V. Biankin [3], David C. Wedge [4,11,13] ✉ & David K. Chang [3,6,13] ✉

Intraductal papillary mucinous neoplasms (IPMNs) are critical precursors to pancreatic ductal adenocarcinoma, a highly lethal cancer characterized by late detection and rapid progression. Here we integrate multi-region whole-genome and transcriptome sequencing to trace the evolution of IPMN, constructing detailed phylogenetic trees to provide insights into subclonal architectures and progression pathways. Our analysis identifies two distinct evolutionary trajectories: one driven by a single ancestral clone, and another involving multiple independent ancestral clones. We further explore the roles of mutational signatures and structural variants in promoting clonal evolution and the emergence of new subclones. Complementing these genomic findings, our transcriptomic analysis reveals unique gene expression profiles and variations in the immune landscape that correlate with different progression stages of IPMN. These insights reveal the complex molecular dynamics of IPMN heterogeneity and progression, highlighting the need to refine early detection and treatment strategies.

Pancreatic ductal adenocarcinoma (PDAC) is associated with a dismal prognosis and is projected to be the second leading cause of cancer related deaths soon[1]. This poor prognosis is largely attributed to the late-stage presentation, which is further compounded by rapid disease progression once diagnosed.

As one of the precursor lesions of pancreatic cancer, intraductal papillary mucinous neoplasm (IPMN) are cystic lesions that are radiologically detectable, offering a unique opportunity for detection and intervention to prevent malignant transformation. Current clinical recommendations for surgical resection of IPMNs are primarily based

on imaging criteria, leading to large numbers of patients undergoing annual imaging surveillance[2]. A better understanding of the critical molecular events that drive progression in premalignant lesions is urgently needed to better manage these patients to achieve early diagnosis and cancer prevention.

Genomic studies on PDAC have advanced our understanding of its molecular landscape[3–6]. However, these studies have often focused on single tumor regions without integrating detailed pathological or radiological data, resulting in cohort analyses that combine PDACs arising from various precursor lesions. Studies modeling progression

from IPMN to PDAC, using single driver mutations, have revealed significant intratumoral heterogeneity, frequently identifying distinct *KRAS* and *GNAS* mutations within the same lesion, particularly in low-grade dysplasia regions[7–10].

This heterogeneity suggests polyclonal origins in IPMNs, contrasting with the clonal homogeneity observed in advanced PDAC[11], reflecting distinct evolutionary patterns underlying early and late stages of tumorigenesis. These findings highlight the need for comprehensive analyses of the genetic complexity within individual lesions, as the contributions of other genomic alterations, such as structural variants (SVs) and copy number alterations (CNAs), remain poorly understood.

This study employs multi-regional whole-genome and transcriptomic sequencing on surgically resected IPMN samples to comprehensively analyse intratumoral heterogeneity, clonal evolution, and associated molecular pathways. By integrating genomic and transcriptomic data, we provide insights into the evolutionary trajectories and signaling networks underlying IPMN progression, advancing our understanding of the pathways leading to pancreatic cancer.

## Results

**Patient characteristics** A cohort of 12 patients underwent pancreatic resection for high-risk IPMNs following clinical guidelines (Supplementary Data 1). Forty-seven tumor samples, plus matching normal tissue from each patient, were harvested from 12 surgical specimens (mean of 4 regions per tumor, range 2–6). Six cases (Cases 6, 9, 7, 4, 15, 16) provided samples representing different grades of dysplasia and histologically associated cancer within the same surgical specimen. Whole-genome sequencing (WGS) was successfully performed on 54 samples (42 tumor samples, mean coverage 80X, and 12 normal samples at 60X). Thirty-seven tumor samples were included in the clonal analysis, excluding 5 with low tumor purity or mutation counts. Transcriptome sequencing (RNAseq; average 100 million paired reads) was performed on 36 tumor samples, 32 of which had matching WGS data (Figure S1A2).

### Mutational landscape of IPMN

A total of 66,724 single-nucleotide variants (SNVs), 4683 insertions/deletions (Indels), and 2447 structural variants (SVs) were identified across all tumor samples. These spanned 48 driver genes and 11,200 other genes across IPMN and associated PDAC samples. Tumor mutation burden (TMB) (4.11 vs 1.38, $p = 0.0001$; Fig. S1B1a, Supplementary Data 2), as well as the mean number of SNVs (3558 vs. 1608, $p = 0.0038$), Indels (253 vs. 105.5, $p = 0.009$), and SVs (91.8 vs. 31.7, $p = 0.0004$), were significantly higher in PDAC compared to IPMN. Higher SV counts were also observed for large translocations, tandem duplications, deletions, and inversions in PDAC (Figs. 1B and S1B1c, Supplementary Data 3).

Key driver mutations were observed in tumor suppressors (*ARID1A, ATM, CDKN2A, LRP1B, NBEA, RNF43, SMAD4, TP53*) and oncogenes (*FAT3, GNAS, KMT2C, KRAS, ZNF521*). *KRAS* was the most frequently mutated gene, with hotspot mutations (15/22 G12D, 5/22 G12V, 2/22 G12R) present in both IPMN and PDAC samples and no significant differences in term of cancer cell fractions (CCFs). Case 4 and 15 showed two distinct *KRAS* variants within the same lesion, suggesting intralesional heterogeneity (Fig. S1B2–3 and Supplementary Data 4). *TP53* SNVs/Indels were predominantly seen in high-grade IPMN and PDAC. *LRP1B* mutations were exclusive to invasive cancer (0 in IPMN vs 7 in PDAC, $p = 0.0056$), while *GNAS* mutations were more frequent in low-grade IPMN (8 in IPMN vs 1 in PDAC, $p = 0.0297$) (Fig. S1B4).

Driver SVs in exonic regions were detected in Case 7 and Case 16, including alterations in *MUC4* in high-grade IPMN and PDAC, *MDM2* in both low- and high-grade lesions, and *MYB, THRAP3*, and *PDCD1LG2* in high-grade IPMN and PDAC (Fig. 1C).

High-frequency CNAs were identified in 64 regions (33 gains, 18 losses, and 13 complex regions). The most frequent regions were

17p11.2 (gain 13/41, loss 12/41, complex 15/41), 20q11.21 gain (36/41), and 9p21.3 loss (25/41). Most CNAs (52/64) persisted from non-invasive IPMN to PDAC (Fig. S1C.A), with four gains and four losses showing higher prevalence in PDAC. Interestingly, 17p11 gain was more common in IPMN than PDAC (Fig. S1C.B and Supplementary Data 5). Losses in 11 driver genes were frequent, including *CDKN2A* (9p21.3), *LAT1S1, ARID1B, SMAD4, U2AF1*, and *RNF43*, with *RNF43* and *U2AF1* losses significantly associated with high-grade lesions ($p = 0.0086$ and $p = 0.0049$ respectively) (Figure S1B4).

### Mutational signatures in IPMN

Mutational signature analysis identified eight single-base substitution (SBS) signatures, with SBS1 and SBS5 (age-related) predominant across all samples. AID/APOBEC-related SBS2 and SBS13 emerged only in clones of Case 7 progressing to cancer. Indel signatures ID1 and ID2, linked to mismatch repair deficiency, were present in cases 12 and 7, respectively, with five additional indel signatures detected. However, all cases retained expression of MMR proteins by immunohistochemistry, and no molecular evidence of microsatellite instability—including low overall tumor mutational burden—was observed, suggesting that ID1 and ID2 may develop through MMR-independent mechanisms[12].

Structural signature SV4 was significantly enriched in PDAC samples ($p = 0.011$). Copy number signature CN9, indicating chromosomal instability, was observed in 31 samples, while a CN48B signature, marked by LOH segments, was identified in 21 samples from 10 cases (Fig. S2A, B). Notably, CN48B activity showed a moderate, statistically significant positive correlation with ID1 and ID2 ($R = 0.53$ and $0.55$; $p = 0.003$ and $0.0002$, respectively), suggesting that CN48B may reflect broader defects in DNA repair or replication fidelity (Fig. S2B, C).

### Heterogeneity in genomic evolution of IPMNs

Mutations were clustered according to their CCFs to construct phylogenetic trees from multiregional samples of 11 cases, representing for each case the genetic relationships among subclones based on the hierarchy of oncogenic events. The analysis revealed two distinct evolutionary trajectories: (1) all IPMNs develop from one single most recent common ancestor (MRCA) ($N = 8$; Cases 3, 6, 7, 9, 10, 12, 13, 16), and (2) IPMNs develop from separate, independent MRCAs ($N = 3$; Cases 2, 4, and 15).

Further heterogeneity in tumor evolution was observed within both single and multiple independent MRCA trajectories. In the single MRCA cases, driver alterations (including SNVs, Indels, and CNAs) occurred clonally and subclonally in both oncogenes and tumor suppressor genes (Cases 3, 6, 7, 9, 10, and 12). In contrast, in Cases 13 and 16, driver alterations were confined to subclones.

Among the multiple independent MRCA cases, Case 15 exhibited IPMN_HGD and PDAC samples initiated from distinct MRCAs, both harboring driver mutations, while other clones without driver mutations remained inactive in subclonal progression. Similar independent evolutionary patterns were observed in Cases 2 and 4.

Across all samples, subclonal progression revealed uneven branch lengths in the phylogenetic trees of Cases 6, 7, 9, and 10. Notably, Case 6 displayed 1,932 mutations in the red branch compared to only 32 in the green branch, suggesting a late subclonal sweep in the red branch (Fig. 2A). These findings highlight that critical genomic events occurred not only at the MRCA level but also in descendant clones, contributing to malignant transformation.

To increase statistical power and validate our findings using external PDAC data, we performed mutational timing analysis on the accessible PDAC samples from the PCAWG dataset using our in-house developed approach[4,13,14]. Although the PCAWG cohort is based on single-region sequencing and includes both conventional PDACs and those arising from IPMNs, the distribution of mutations across timing categories, particularly the distinction between clonal early and subclonal mutations, supports our key observation of pervasive clonal

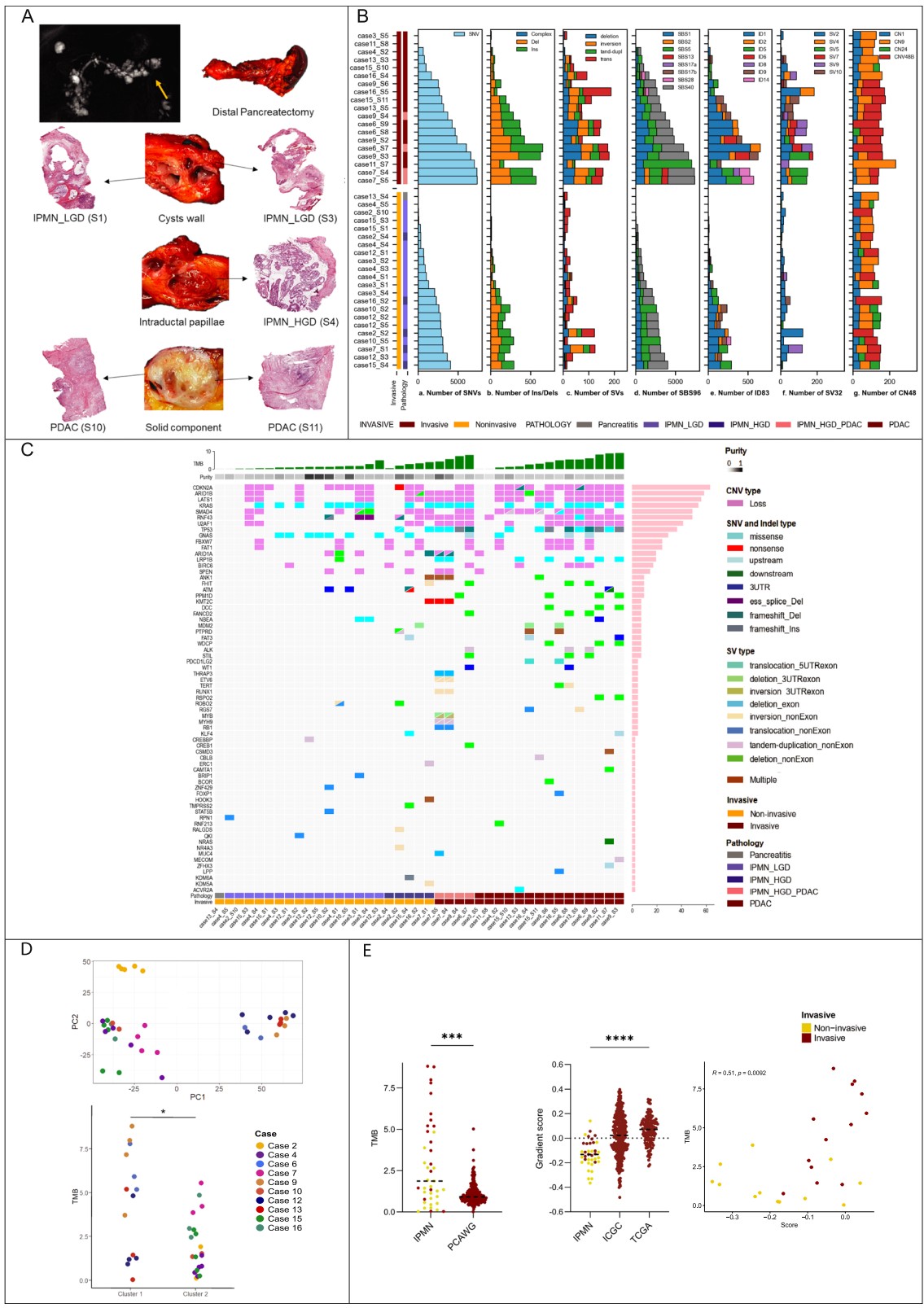

heterogeneity across all samples (Fig. S3.A). When comparing pooled proportions of clonal early versus subclonal mutations, no statistically significant difference was observed (Fig. 3S.B). However, four recurrent driver mutations, most notably KRAS and TP53, were predominantly clonal early, consistent with their role as early trunk events. In contrast, mutations in SMAD4 and LRP1B occurred both early and late, supporting a model of subclonal evolution (Fig. 3S.C).

These findings align with our current study (Fig. 2.A), reinforcing the temporally ordered acquisition of driver mutations in PDAC.

## Single vs multiple branch clonal evolution in IPMN

By investigating whether lineages in different patients are associated with specific genomic alterations, we identified two patterns of subclone expansion: (1) a linear progression where the ancestral clone gives rise to

**Fig. 1 | Mutational landscape of IPMN. A Multiregional Sampling Protocol.**
Representative example from Case 15 illustrating the multiregional sampling approach designed to capture different areas within a single lesion. Preoperative MRI for Case 15 showed a multifocal IPMN in the pancreas with a solid component in the tail (arrow in yellow), prompting distal pancreatectomy. Fresh-frozen tissue samples were collected for molecular analysis, including IPMN_LGD (low-grade dysplasia) from the cyst walls, IPMN_HGD (high-grade dysplasia) from intraductal papillae, and PDAC (pancreatic ductal adenocarcinoma) from the adjacent solid component. **B Mutational landscape of IPMNs and PDACs.** Total number of SNVs (a), INDELS (b), and SVs (c), including deletions, inversions, tandem duplications, and translocations, for each sample grouped as IPMN (lower) and PDAC (upper). PDAC samples exhibited significantly higher average counts of SNVs, Indels, and each breakpoint type compared to IPMN ($p < 0.05$ for all genomic alterations). Proportions of single-base substitution (SBS) (d), Indel (ID83) (e), and copy number (CN48) (f) COSMIC signatures are shown for each sample. A total of 7 Indel (ID) signatures, including ID1, 2, 5, 6, 8, 9, and 14, were identified across 41 samples. Additionally, 4 CN signatures, including CN1, CN9, CN24, and CNV48B, were

observed across the cohort. **C Key genomic drivers in IPMN progression:** Driver SNVs, Indels, SVs, and CNAs are highlighted, illustrating the progression from low-grade IPMN to high-grade IPMN and invasive PDAC. **D PCA of gene expression, revealing two distinct clusters.** Cluster 1 (cases 6, 9, 12, 13) and Cluster 2 (cases 2, 4, 7, 10, 15, 16). Cluster 1 exhibited increased SNV-indel TMB compared to Cluster 2 (mean 4.33 vs 2.13; $p = 0.03$, two-sided Fisher–Pitman permutation test). **E TMB and the transcriptomic profiles comparison between IPMN and distinct PDAC cohorts. a.** TMB comparison between the IPMN cohort (41 samples) and the PCAWG cohort (241 samples), $p = 0.0001$, two-sided Mann–Whitney test. **b.** Distribution of transcriptomic gradient scores (classical vs. squamous) in the 37 study samples, compared to 362 ICGC and 149 TCGA cases, with higher scores indicating squamous tumor characteristics, $p < 0.0001$, two-sided Mann–Whitney test. **c.** Scatter plot showing the correlation between TMB and the gradient score within the IPMN cohort. Lower transcriptomic scores in IPMNs suggest an earlier disease stage, with a trend towards higher TMB and increasing squamous characteristics as the disease progresses. Source data are provided as a Source Data file.

a single subclone, and (2) a branching progression where the ancestral clone generates multiple subclones diverging independently.

In the linear model (Cases 4, 12, 13, and 15), a dominant clone acquires successive alterations, generating a sequential series of subclones, each retaining the genetic features of its predecessor while gaining unique mutations. This stepwise progression highlights a hierarchical evolution of disease. In contrast, the branching model (Cases 2, 3, 6, 7, 9, 10, and 16) involves multiple subclones diverging independently from the ancestral clone, resulting in a complex architecture with distinct lineages, each harboring unique genetic alterations in addition to those inherited from the original clone.

While these patterns are distinct, the observed tree shapes may be influenced by random sampling of IPMN regions. Notably, cases with branching evolution exhibited significantly higher counts of intrachromosomal structural variations (median 85.2 vs 29.7, $p = 0.0024$), suggesting a potential role for SVs in driving the generation of diverging subclones (Fig. 2B, C).

## Temporal dynamics of mutational signatures

To study the mutational processes during clonal progression in IPMNs, we calculated the proportion of mutations assigned to each mutational signature across the branches of the tumor phylogenies (Fig. 2D). Overall, SBS1, SBS5 (clock-like/age-related), and SBS40 were the most prevalent signatures, present in both the trunk and branches of the phylogenies. The AID/APOBEC-associated signatures SBS2 and SBS13 emerged in the MRCA of Case 7 but were restricted to the branch associated with invasive PDAC. Similarly, in Case 16, SBS28 was observed only in the PDAC subclone, and not in the pre-malignant IPMN branch.

ID1 and ID2 signatures, linked to DNA mismatch repair deficiency, were observed in the early stages of tumor initiation in most cases (2, 3, 4, 6, 7, 9, 15, 16) but were unexpectedly absent in subclones. A similar trend was noted for other indel signatures (ID5, ID6, and ID9), which appeared in earlier clones but not in subclones. This discrepancy may reflect the smaller number of mutations assigned to subclones compared to trunks and the overall lower prevalence of indels relative to SNVs, potentially limiting the detection of certain signatures due to insufficient power.

## Modeling malignant transformation from IPMN to PDAC

In six cases with distinct grades of dysplasia and/or associated cancer, two (Cases 6 and 9) included IPMN with microinvasive PDAC (IPMN_HGD_PDAC), while four (Cases 7, 4, 15, and 16) provided samples of both IPMN and invasive cancer. In Case 2, WGS was performed on both IPMN and PDAC samples, but the analysis was successful only for the IPMN.

In the phylogenetic tree of these cases, the branch point where IPMN and PDAC diverge indicates the emergence of infiltrative clones.

In Case 9, the clonal architecture showed a branching pattern, with the ancestral clone (gray) giving rise to several independent subclones (Fig. 2A). The MRCA clone, carrying *KRAS*, *TP53*, and *LRP1B* mutations, as well as the "blue" and "green" subclones were observed in both IPMN_HGD_PDAC and PDAC samples, suggesting their potential to progress to invasive cancer. In contrast, subclones found only in PDAC samples ("yellow, pink, and purple") likely represent a secondary wave of clonal expansion specific to cancer. Notably, one PDAC sample (S3) had multiple intrachromosomal SVs and contained most subclones, suggesting that these SVs may have played a role in driving subclone development.

Similarly, in Cases 6 and 16, a single MRCA was shared by IPMN_HGD and PDAC samples, suggesting that the MRCA in HGD may have already acquired invasive potential. In Case 6, similar SV patterns were detected in both HGD and PDAC samples. In contrast, in Case 16, PDAC samples (S4 and S5) showed a higher number of intrachromosomal SVs than the HGD sample (S2), potentially driving invasive disease and PDAC-specific subclones (green).

In Cases 4 and 15, distinct ancestral clones were identified for IPMN and invasive cancer samples. Notably, in Case 15, HGD clones carried KRAS, GNAS, and ATM mutations, while invasive cancer clones displayed different mutations in KRAS and TP53 (Fig. 2A), indicating separate evolutionary trajectories for high-grade lesions within the same patient.

## Transcriptomic patterns in IPMNs

To explore the biological programs associated with IPMN progression, we examined gene expression dynamics across individual cases (Fig. S4). Although some pathways showed differential enrichment between invasive and non-invasive samples, only adipogenesis, androgen response, and heme metabolism were significantly more enriched in non-invasive samples, consistent with metabolic reprogramming during progression (Fig. S5A, B).

Principal Component Analysis (PCA) of RNAseq data from samples matched to WGS revealed two main transcriptomic clusters: Cluster 1 (Cases 6, 9, 12, 13) and Cluster 2 (Cases 2, 4, 7, 10, 15, 16) (Fig. 1D). Although histological features were comparable between the clusters, Cluster 1 was genomically enriched for LRP1B mutations, exhibited higher TMB (SNVs + indels, $p = 0.02$), and displayed lower expression of most cancer hallmarks, except for those related to development, bile acid metabolism, and KRAS signaling (downregulated target enrichment) (Fig. 3A–C)[15].

Principal Component Analysis (PCA) of RNAseq data from samples matched to WGS identified two distinct gene expression clusters: Cluster 1 (Cases 6, 9, 12, 13) and Cluster 2 (Cases 2, 4, 7, 10, 15, 16) (Fig. 1D). Cluster 1 showed lower expression of most cancer hallmarks, except those related to development, bile acid metabolism, and downregulated KRAS signaling[15], genomically, higher TMB (combining

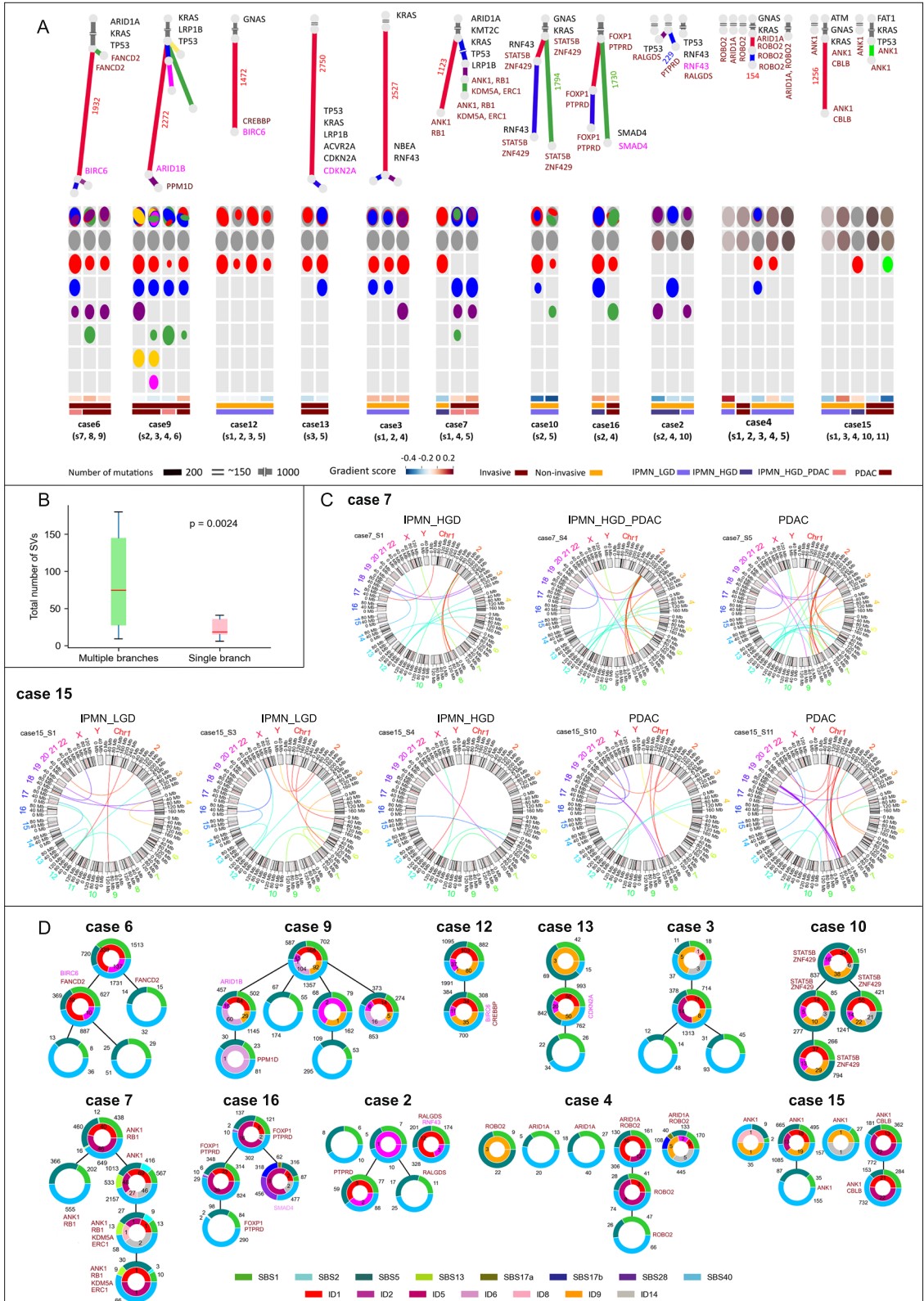

SNV and indels, $p = 0.02$) and was enriched for LRP1B mutations, with no discernible histological differences (Fig. 3A–C).

Following gene expression normalization, a transcriptomic gradient score was assigned to each sample and compared to reference PDAC cohorts (ICGC PACA-AU[16] and TCGA-PAAD cohorts[14]). Higher scores indicated greater alignment with the squamous molecular subtype (Figs. 2A and S6).

IPMNs showed overall lower gradient scores compared to PDACs, indicating a higher prevalence of the Classical subtype, reflective of earlier-stage, progenitor-like disease with better prognosis (Fig. 1E). Classical subtype uniformity was observed in non-invasive IPMN cases (2, 10, 12), while significant intratumoral variation was evident in Case 4, where an LGD sample (S1) exhibited a squamous-like signature alongside increased SVs

**Fig. 2 | Patterns of genomic heterogeneity and evolution in 11 cases of IPMNs. A Subclone structure in IPMNs and PDACs.** All the subclones identified in the WGS samples are reconstructed as phylogenetic trees (Upper) and oval plots (Lower). Phylogenetic trees show the relationships between subclones in the samples of each patient. Branch lengths are proportional to the number of SNVs and INDELs in each cluster. Branch annotations include the samples containing the subclone and genetic alterations, such as SNVs/Indels (black), SVs (brown), and CNAs (pink). The bottom legend shows the length scale in total SNVs and indels (-150, 200, and 1000 mutations). Mutation counts are labeled on the longest branch of each tree in the same color as the branch. On the left, eight cases share a common MRCA. On the right, three cases originate from multiple independent tumors. Oval plots represent subclone composition in each sample, with subclone areas proportional to their CCF (Supplementary Data 6). **B Impact of Structural Variations on clonal progression.** Cases with branching clonal evolution ($n = 18$ spatially distinct regions) exhibited significantly more structural variants (SVs) compared to single-branch cases ($n = 16$; mean 85.2 vs 29.7; $p = 0.0024$, two-sided Fisher–Pitman permutation test), emphasizing their contribution to genomic heterogeneity and tumor progression. Box plots display the median (centre line), the interquartile range (box), and the minima and maxima (whiskers). **C Structural Variations in Clonal Evolution of IPMN Cases.** In Case 7 (single-branch clonal evolution), increased SVs were observed in PDAC compared to HGD, alongside new SVs in HGD, indicating ongoing evolution after the appearance of the infiltrating clone. In Case 15 (multiple ancestral clones), distinct SV patterns in samples derived from independent ancestral clones highlight the independent evolution of these lesions. **D Mutational Signatures Across Clonal and Subclonal Lineages.** Phylogenetic trees with unscaled branches show mutational signatures annotated in pie charts for each branch. Common signatures (SBS1, 5, 40) appear in clones and subclones of both single and independent MRCA cases. Early-stage signatures (ID1, ID2) were present in tumor initiation but absent in later subclones in several cases (2, 3, 4, 6, 7, 9, 15, 16). Branch annotations indicate samples with driver SVs (brown) and CNAs (pink). Source data are provided as a Source data file.

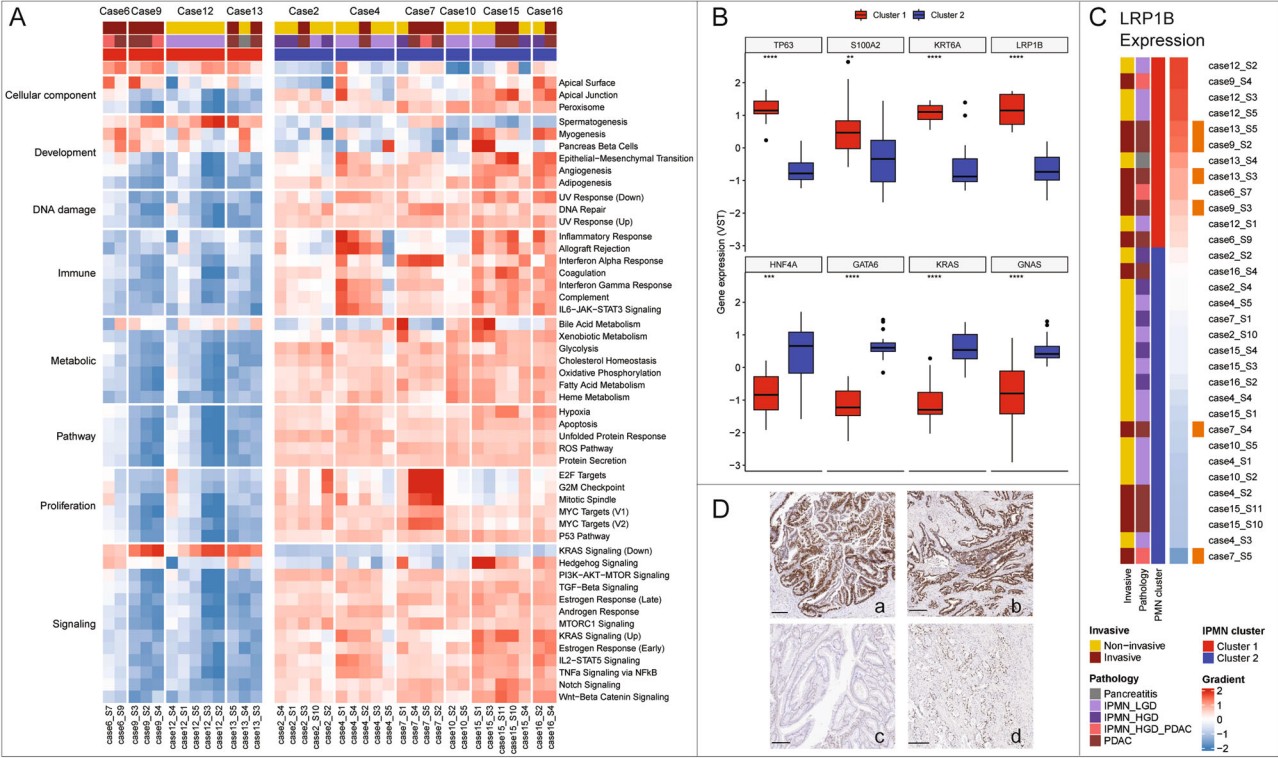

**Fig. 3 | Gene and protein expression profiles across transcriptomic clusters. A Heatmap of cancer-related pathway enrichment per sample,** with Cluster 1 samples on the left and Cluster 2 on the right, reflecting the PCA analysis from Fig. 1. **B Differential gene expression between the clusters.** TP63 ($p < 0.0001$), S100A2 ($p < 0.01$), KRT6A ($p < 0.0001$), and LRP1B ($p < 0.0001$) are upregulated in Cluster 1 ($n = 13$ distinct tumor regions), while HNF4A ($p < 0.001$), GATA6 ($p < 0.0001$), KRAS ($p < 0.0001$), and GNAS ($p < 0.001$) are higher in Cluster 2 ($n = 24$), two-sided unpaired $t$-test. Box plots display the median (centre line), the interquartile range (box), and the minima and maxima (whiskers). **C Correlation of LRP1B expression levels with SNVs.** Cluster 1 samples (top) show elevated LRP1B expression and SNVs, while Cluster 2 samples (bottom) display reduced expression. **D P53 immunohistochemistry on diagnostic FFPE samples (scale bar: 250 μm).** Case 7 demonstrates P53 mutation in both IPMN (**a**) and PDAC (**b**), while Case 15 shows normal P53 expression in IPMN (**c**) and overexpression in PDAC (**d**), validating the genomic findings. Source data are provided as a Source data file.

and CNAs compared to other samples from the same tumor (Fig. 2A).

Although the two clusters exhibited similar transcriptomic gradient scores overall—with greater variability in Cluster 2—distinct expression patterns were observed. Cluster 1 displayed a transcriptional profile consistent with squamous subtype differentiation, marked by elevated expression of squamous-associated genes (TP63, KRT6A, S100A2) and reduced expression of classical transcription factors (GATA6, HNF4A) (Fig. 3A, B). In contrast, Cluster 2 reflected a more classical/progenitor phenotype, characterized by higher expression of KRAS, GNAS, GATA6, and HNF4A, suggestive of a more

differentiated transcriptional state and potentially less aggressive biological behavior[16–18].

**Tumor microenvironment analysis**

Using EPIC cell proportion estimates[19], we assessed the tumor microenvironment (TME) during IPMN progression (Fig. 4A). Cluster 1 demonstrated higher CD8+ T cell populations compared to Cluster 2 (0.08 vs 0.01, $p = 0.01$), despite its higher squamous subtype scores, while Cluster 2 exhibited increased macrophage populations (0.006 vs 0.009, $p = 0.04$) (Fig. 4B). PDAC samples showed significantly elevated CAF populations (median 0.24 vs 0.04, $p = 0.04$) and reduced

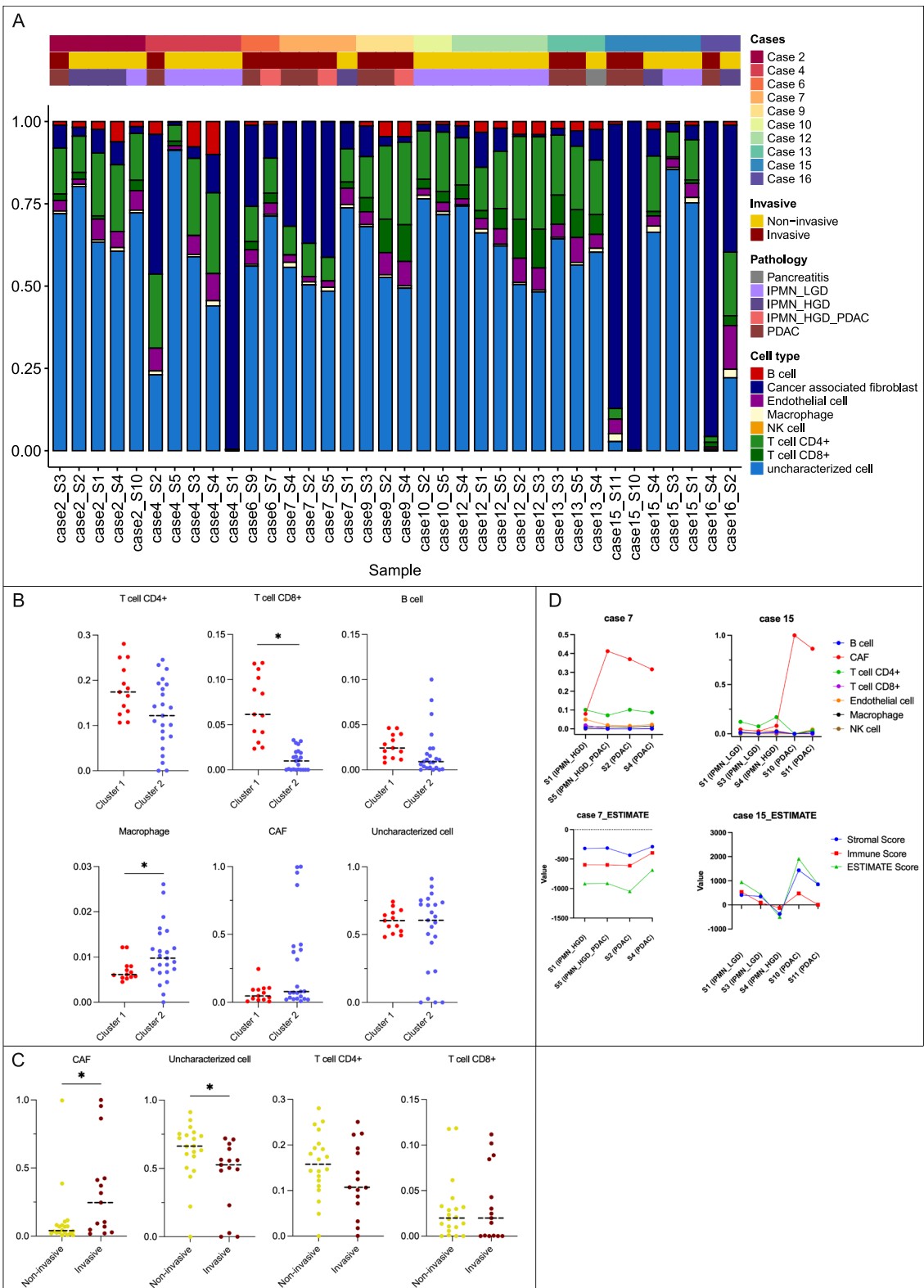

**Fig. 4 | Cellular composition dynamics in IPMN progression. A** EPIC cell proportion estimates for each sample, detailing the cellular composition across cases. **B Comparison of cell populations between transcriptomic clusters**. Cluster 1 shows higher CD8[+] T-cell proportions (median 0.08 vs 0.01, $p = 0.02$), while Cluster 2 has increased macrophage populations (median 0.006 vs 0.01, $p = 0.01$). Comparisons were performed using the median cell proportion per case, derived from 13 spatially distinct tumor regions in Cluster 1 and 24 in Cluster 2 (two-sided

Mann−Whitney test). **C** Cellular differences between IPMN ($n = 20$) and PDAC samples ($n = 15$) highlight increased CAF populations (median 0.04 vs 0.24, $p = 0.02$) and reduced uncharacterized epithelial cells (median 0.66 vs 0.52, $p = 0.01$), two-sided Mann−Whitney test, consistent with the transition from non-invasive to invasive stages. **D** Representative cases (Cases 7 and 15) show immune cell composition changes and ESTIMATE-derived scores tracking progression from non-invasive IPMN to PDAC. Source data are provided as a Source data file.

epithelial cell populations (median 0.52 vs 0.66, p = 0.01) compared to premalignant IPMNs, indicating a stromal shift associated with invasive disease (Fig. 4C, D). CD4+ and CD8+ T cell counts were generally lower in PDAC compared to IPMN across cases (7, 9, 15, 16), though these differences did not reach statistical significance.

In LGD_IPMN samples (Cases 10, 12, 15), cellular composition remained consistent, characterized by higher CD4+ T cells and reduced CAFs compared to high-grade lesions. ESTIMATE scores[20], which summarize stromal and immune components, confirmed positive stromal scores in most PDAC samples, with the exception of Case 7, where histology revealed prominent inflammatory infiltration. Higher stromal scores were associated with increased CAF populations and squamous subtype enrichment (Fig. S7, S8, S9). Correlation analysis revealed that higher Squamous Scores were moderately associated with increased ESTIMATE Scores, strongly associated with stromal scores, suggesting that squamous features in IPMNs are linked to a more stromal-rich tumor microenvironment. This pattern was validated in the ICGC cohort, where the squamous subtype showed elevated stromal and ESTIMATE scores, while immune infiltration remained weakly associated (Fig. S8).

TME cell proportions were compared between two distinct evolutionary trajectories: single MRCA vs multiple independent clones pathway. No significant differences were observed in B cells, macrophages, or CD4$^+$ T cells between the two trajectories. However, CD8$^+$ T cell proportions and ESTIMATE Immune scores were significantly elevated in tumors that evolved through the single MRCA pathway (Fig. S10).

## Discussion

Here, we present a comprehensive exploratory analysis of WGS and RNAseq from multi-regional samples of IPMNs collected from patients undergoing surgical pancreatic resection. Using fresh-frozen tissue, we performed an in-depth genomic analysis to explore the subclonal architecture. This analysis was integrated with mutational signatures, copy number alterations (CNAs), and structural variants, providing insights into the molecular mechanisms underlying neoplastic progression and intratumoral heterogeneity of IPMNs. Various classes of somatic mutations were observed in coding or non-coding regions in both IPMN and PDAC, with PDAC displaying a significantly higher prevalence of both simple and complex structural variations. Despite IPMNs and PDAC sharing critical genomic alterations, such as KRAS mutations and CDKN2A losses, high-grade lesions were enriched for SNVs and indels in TP53 and LRP1B, along with losses in RNF43 and U2AF1, while GNAS mutations were predominantly found in low-grade lesions, consistent with prior studies[21,22]. These findings highlight the stepwise accumulation of genomic alterations, including driver mutations and intricate chromosomal rearrangements, during neoplastic progression[23]. In our study, RNF43 alterations included both SNVs and copy number losses, with the latter significantly enriched in high-grade lesions—an event not detectable in previous targeted or WES-based studies[8,9,22]. KLF4 mutations, predominantly described in LGD lesions[24], were identified in one HGD and one PDAC sample. Given our sampling strategy focused on macroscopically visible lesions, early KLF4 alterations in smaller LGD foci may have been underrepresented.

We observed two distinct evolutionary patterns: one driven by a single ancestral clone and another characterized by parallel emergence of independent genomic lesions within the same tumor. Low-mutational-burden IPMN samples were indicative of early onset, whereas the acquisition of driver mutations promoted one clonal trajectory over others, leading to further genomic aberrations and the development of subclones. Notably, cases 4 and 15 exhibited distinct mutations in identical driver genes, consistent with findings from previous exome and targeted sequencing studies[7–9,22]. However, prior progression models, showing low-grade IPMNs with distinct KRAS mutations, suggested early clonal heterogeneity with progression to high-grade lesions via clonal selection[8,9,25]. Our findings from case 15,

where both cancer and adjacent IPMN_HGD displayed distinct genomic alterations, suggest that polyclonal evolution in IPMNs may persist even at more advanced stages.

While these studies laid foundational insights into IPMN evolution, our application of multi-regional WGS has enabled the detection of structural variants and non-coding alterations, providing enhanced resolution into clonal independence and the genomic dynamics of progression. This approach has revealed the absence of shared SVs and CNAs across genomically distinct regions within the same lesion, supporting a model of parallel evolution and raising the possibility that non-genomic alterations may contribute to tumor initiation within the pancreatic ducts. Although sampling was carefully focused on distinct areas of the same lesion (e.g., cyst wall, papillae, solid component), histological confirmation could not exclude the possibility of microscopic lesions such as PanINs deeper within the specimen, potentially contributing to the observed genetic independence among clones[26]. Conversely, a single MRCA initiated a lineage with either one (Cases 3, 13, and 12) or more subclones (Cases 7, 9, 10, and 16), leading to the formation of HGD or PDAC through the accumulation of additional genetic alterations. Interestingly, subclones identified in IPMN_HGD samples were absent in the corresponding PDAC, suggesting parallel and independent genomic evolution of HGD and PDAC despite originating from the same genetic lineage. This aligns with observations from clinical studies on resected IPMNs, where resection of HGD confers survival benefits, even when cancer is also resected, suggesting that HGD progresses through a divergent pathway alongside invasive components[7,27].

To investigate whether immune contexture differed between evolutionary trajectories, we compared tumor-infiltrating immune cells across clonal patterns. Recent IHC studies consistently reported higher CD8$^+$ infiltration in LGD than in advanced IPMNs, suggesting that the early immune microenvironment is generally more permissive to T-cell infiltration, irrespective of underlying genomic heterogeneity[28,29]. In our study, CD8$^+$ cytotoxic T-cell infiltration and Immune Scores were higher in tumors evolving from a single MRCA, indicating that clonal architecture during progression further modulates this effect. Although we did not directly assess neoantigen presentation, the presence of multiple independent clones is likely to generate distinct, lineage-specific alterations that fragment and dilute antigenic visibility, thereby contributing to immune evasion. This interpretation is consistent with prior studies showing that tumors enriched for clonal (trunk) rather than heterogeneous neoantigens are associated with stronger immune responses[30–32].

Cancer genomes evolve through dynamic mutational processes, accumulating alterations over time and leaving specific mutational signatures[33,34].

In this study, the most common SNV signatures—SBS1, SBS5 (clock-like/age-related), and SBS40—were consistently detected across both trunk and branch mutations, supporting their role in IPMN initiation and progression.

APOBEC-associated signatures (SBS13 and SBS2) were found exclusively in the PDAC subclone of Case 7 and not in the HGD-specific subclone, suggesting that APOBEC mutagenesis may contribute to genomic instability specifically during malignant progression.

By contrast, indel signatures ID1 and ID2—typically linked to mismatch repair deficiency—were observed in early clones. However, all cases demonstrated intact MMR protein expression and no evidence of microsatellite instability, indicating these signatures likely arise through alternative, MMR-independent mechanisms[35–37]. Together, these findings support a model in which early IPMN genomes are shaped by age-related and replication-linked processes, with APOBEC activity and chromosomal instability emerging later during malignant progression.

CNA and SV signatures were also critical in tumor evolution, with increased CNAs and SVs observed in PDAC samples. We identified a copy number signature, CNV48B, characterized by extensive LOH and

associated with ID1 and ID2 activity, suggesting a potential link to underlying chromosomal instability. However, its etiology and clinical relevance require further validation in larger cohorts.

Notably, cases with branching clonal evolution exhibited a significantly higher number of SVs, underscoring their role in subclonal formation and divergence[4,16]. This observation suggests that SVs contribute to intratumoral genomic evolution and challenge previous progression and timing models based solely on SNVs, which may not capture the full spectrum of genomic instability in IPMNs[8].

In line with PDAC transcriptomic subtypes[16-18], our clustering analysis revealed biologically meaningful divergence among IPMN samples. Despite overall similar squamous scores, Cluster 1, characterized by immune-enriched features and elevated TMB, exhibited increased expression of genes associated with squamous differentiation—a phenotype linked to therapeutic resistance and poor prognosis in PDAC[20,29]. This transcriptional profile, accompanied by elevated LRP1B expression and reduced WNT signaling activity, suggests a coordinated shift toward immune adaptation and epigenetic reprogramming during malignant transformation[29,30].

LRP1B, a gene associated with the WNT pathway and frequently mutated in lung and gastrointestinal cancers, has been linked to higher TMB, increased immune infiltration, and improved immunotherapy responses[38-40], however, its role in pancreatic cancer remains to be clarified. In contrast, Cluster 2 retained a more classical/progenitor-like phenotype, suggestive of a transcriptionally stable trajectory and associated to better oncological outcomes. These findings support the hypothesis that early transcriptional divergence may underlie distinct evolutionary and clinical trajectories in IPMNs, reinforcing the potential utility of transcriptomic subtyping for future risk stratification.

These findings provide a deeper understanding of the molecular complexity underlying IPMN progression and highlight the need for prospective studies that address this heterogeneity. While cyst fluid analysis holds promise for IPMN management, its limitations in detecting driver mutations highlight the need for improved methodologies[9]. Our integrated multiregional WGS and RNAseq approach offers a foundation for developing strategies to predict IPMN progression and refine tools for cancer prognostication.

Study limitations include the following. First, the small number of patients, despite multi-regional sampling, may not fully capture the spectrum of IPMN molecular variations. IPMNs are indeed characterized by vast heterogeneity in morphology and clinical behavior, which may reflect varied patterns of cancer evolution and progression. The sample size, however, reflects the comprehensive nature of the analysis, which required significant resources to examine each sample at high resolution.

Second, the cohort consisted entirely of surgically resected IPMNs, introducing selection bias as these lesions were identified based on radiologic criteria warranting resection. This limits the ability to compare findings with the majority of IPMNs managed through surveillance and not progressing to cancer.

Lastly, our reliance on frozen tissue, essential for performing WGS, does not allow for the spatial localization of different genetic clones within FFPE tissue. Gaining such spatial context could offer additional understanding of the interaction dynamics between tumor clones and immune cells, improving our comprehension of tumor behavior and progression pathways.

In conclusion, this study advances understanding of IPMNs progression by describing distinct patterns of clonal expansion and confirming their polyclonal origin. We demonstrate the genomic independence of distinct clones at a deeper level and highlight the critical role of structural variations (SVs) in driving tumor evolution. Transcriptomic analysis revealed distinct clusters and gene expression patterns associated with progression, underscoring the molecular complexity of IPMN evolution. Future studies integrating transcriptomic and whole-genome approaches, particularly focusing on cyst fluid analysis, could build upon these findings to advance biomarker development and refine strategies for IPMN management.

## Methods

### Patient cohorts and multiregional samples collection
Primary tumor samples were collected from patients undergoing surgery at Glasgow Royal Infirmary, UK, and the Pancreas Institute, University of Verona, Italy, following clinical guidelines for IPMN management[2,41,42]. Fresh-frozen tissue was obtained for genomic analysis using a multiregional sampling protocol, dividing each lesion into 4–9 segments, with their positions recorded in a clockwise numeric order for spatial reconstruction. Expert pancreatic pathologists (FD, CL, AS) ensured samples were from the same lesion with no normal tissue in between, to avoid the inclusion of concomitant but separate tumors. All slides from fresh-frozen specimens were centrally reviewed to confirm and classify IPMN or PDAC according to morphology and assess molecular analysis suitability. Following DNA and RNA extraction and quantification from each tumor and matched normal sample, whole genome- (WGS) and RNA- sequencing were performed for comprehensive genomic and transcriptomic analysis. Specific immunohistochemical staining for P53 and SMAD4 was performed on FFPE diagnostic slides for validation of the genomic results. The materials used have been collected under Ethics Committee Approval (ECA) of both institutions (Glasgow: ref. 22/WS/0020; Verona: program 1885 protocol 52438 and amended with protocol 25982).

### Read alignment and somatic variant calling
Paired sequence reads were aligned to the human reference genome (hg38). Somatic single nucleotide variants (SNVs) and insertions/deletions (InDels) were identified by comparing normal and tumor pairs, with filtering applied to generate high-confidence variant datasets. Structural variants were processed and categorized into different types for further analysis.

### Detection of clonal and subclonal copy number alterations (CNAs)
Battenberg algorithm based-pipeline was employed to identify copy number and estimate tumor purity and ploidy as previously described[43] (v2.2.9; https://github.com/Wedge-lab/battenberg). Among the CNAs found in the set of samples, regions altered at a significant high frequency were identified using GISTIC2[44] with the following modified parameters: -brlen 0.7 and -conf 0.99.

### Mutation and copy number signature profiling
Mutation signatures were analysed by using SigProfilerExtractor based on a non-negative matrix factorization (NNMF) framework (v0.0.5.77). Signatures from De novo extraction and decomposition were profiled as single-base substitution (SBS96), double-base substitution (DBS78), and small insertion-deletion (ID83)[36]. Using ASCAT matrix generated from CNA datasets, copy number signatures were extracted by SigProfilerExtractor[45]. Based on the updated known signatures as in COSMIC database, SigProfilerAssignment was utilised to retrieve decomposed signatures.

In order to minimize NNMF artefact potentials, initial signature extraction was performed with all available samples simultaneously[46] and each single signature assignment per sample was finally determined by the probability matrix.

### Mutation clustering and phylogenetic tree reconstruction
Subclonal structures were modeled by analyzing mutation allele fractions, copy number variations, and cellularity data. A previously described Bayesian Dirichlet process (DPClustering)[47], (v2.2.8; https://github.com/Wedge-lab/dpclust/releases) was used to cluster mutations and reconstruct phylogenetic trees, applying principles to determine linear and branching relationships among subclones.

## Annotation of the trees with mutations and signatures

To annotate each tree with oncogenic or putative oncogenic alterations, in-house programs were utilised to connect multiple genomic variants and identify specific significance, including SNV, Indels, SV, CNA, mutation signature, and cluster assignment information from mutation caller, Battenberg, SigProfilerExtractor, and multisample DPClust. All pipelines applied for WGS data analysis, and their steps are illustrated in the flowchart (Fig. S1A2), with detailed descriptions provided in the Supplementary Methods.

## RNAseq analysis

Scores were calculated for each sample for the following transcriptomic subtypes: Bailey subtypes[16], Collisson subtypes[48], and Moffitt subtypes[49]. Two further subtype scores, "Squamous score" and 'Classical score', were calculated for each IPMN sample using a signature derived from RNAseq data from ICGC PACA-AU[16]. A transcriptomic gradient score was therefore obtained by subtracting the classical score from the squamous score. Samples were also scored for ten previously defined gene programs associated with Bailey subtypes and for 50 Hallmark gene sets[15].

Cell type proportions were estimated using the EPIC method as implemented in the immunedeconv[19] package using TPM counts. The ESTIMATE[20] method was used to calculate stromal, immune, and overall ESTIMATE scores, implemented in the estimate (https://bioinformatics.mdanderson.org/ estimate/rpackage.html) package.

## Statistics and reproducibility

All neoplastic and normal samples were sequenced once: whole-genome sequencing was performed for all neoplastic samples, and RNA sequencing for selected cases as described. No statistical method was used to predetermine sample size. No data were excluded from the analyses. The experiments were not randomized, and analyses of genomic and transcriptomic data were conducted independently.

## Reporting summary

Further information on research design is available in the Nature Portfolio Reporting Summary linked to this article.

# Data availability

Whole-genome sequencing (WGS) and RNA-seq raw data have been deposited in the European Genome-phenome Archive (EGA) under accession numbers EGAS50000001182 and EGAS50000001540 respectively. Processed datasets supporting the findings of this study are available at the following repositories: https://github.com/Wedge-lab/IPMNPDACpaperArchive/tree/main/IPMNPDAC_WGS/Data, https://github.com/Wedge-lab/IPMNPDACpaperArchive/tree/main/IPMNPDAC_WGS/Data/sigDPC, https://github.com/Wedge-lab/IPMNPDACpaperArchive/tree/main/IPMNPDAC_WGS/Data/svDriverCluster, https://github.com/Wedge-lab/IPMNPDACpaperArchive/tree/main/IPMNPDAC_WGS/Data/cnvDriverCluster Source data are provided with this paper.

# Code availability

The Code and data used for the analysis are available at: https://gitfront.io/r/xtgitfhe2/8f2rTdyZs1S9/IPMNPDACpaperArchive/ [gitfront.io].

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

## Acknowledgements

This work was performed under the frame of the Scottish Genome Partnership. This study was funded by the Scottish Genome Partnership and, in part, by Associazione Italiana Ricerca Cancro (AIRC IG n. 26343), Italian Ministry of Health through Fondazione Italiana Malattie Pancreas (FIMP_CUP J37G22000230001), Fondazione Cariverona: Oncology Biobank Project "Antonio Schiavi" (prot. 203885/2017), European Union - NextGenerationEU through the Italian Ministry of University and Research under PNRR - M4C2-I1.3 Project PE_00000019 "HEAL ITALIA" CUP: B33C22001030006, European Union - NextGenerationEU through Italian Ministry of Health, PNRR HUB Diagnostica Avanzata PNC-E3-2022-23683266 PNCHLS-DA. DCW carried out this research at the National Institute for Health and Care Research (NIHR) Manchester Biomedical Research Centre (BRC) (NIHR203308). XH was funded by Cancer Research UK RadNet Manchester (C1994/A28701).

## Author contributions

A.P. and D.K.C. designed the study. A.P., R.U.G., C.L., S.D., F.D., N.B.J., C.J.M., E.D., S.R., P.P., N.S., R.T.L., R.S., and A.S. analyzed tissue samples. A.P., R.U.G., C.L., S.D., F.D., N.B.J., C.J.M., E.D., S.R., P.P., N.S., R.T.L., R.S., A.S., and D.K.C. acquired data. A.P., X.H., R.U.G., C.L., L.P.S.S., A.V.B., D.C.W., and D.K.C. analyzed data. A.P., X.H., R.U.G., C.L., L.P.S.S., A.PU., F.E.M.F., M.M., R.S., A.S., A.V.B., D.C.W., and D.K.C. interpreted data. A.P., X.H., D.C.W., and D.K.C. wrote the paper. All authors approved the final version of the manuscript.

## Competing interests

D.K.C. has received consulting and lecture fees from Immodulon Therapeutics and Mylan, research funding from Astrazeneca, BMS GmbH & Co. KG, Merck, and Immodulon Therapeutics. MM reports honoraria from AstraZeneca, MSD Oncology, Ipsen, Hippocrates, Viatris, and Servier, and reports consulting or advisory role for AstraZeneca, MSD Oncology, and Janssen Oncology; research funding from Roche and other relationships (participation to protocol Steering Committees and Independent Data Monitoring Committee) with Novartis and OncoSil. FEMF has received research funding from Astrazeneca and Sierra Oncology, speakers fees from Servier Oncology, Viatris travel support from Viatris, and is in the advisory board of Astellas and Abbott. NBJ has received honorariums and delivered talks for Nanostring, 10x genomics, APkoya, and is the scientific advisory board for Galvani. C.L. has received funds from NTP (scientific advisor), MSD (speaker bureau), and Medica srl (speaker bureau). No potential conflicts of interest were disclosed by the other authors.

## Additional information

[1]Unit of Pancreatic Surgery, University of Verona Hospital Trust, Verona, Italy. [2]Department of Engineering for Innovation Medicine, University of Verona Hospital Trust, Verona, Italy. [3]School of Cancer Sciences, University of Glasgow, Glasgow, Scotland, UK. [4]Manchester Cancer Research Centre, Manchester, UK. [5]Department of Diagnostics and Public Health, Section of Pathology, University of Verona Hospital Trust, Verona, Italy. [6]West of Scotland HPB Unit, Glasgow Royal Infirmary, Glasgow, Scotland, UK. [7]Department of Pathology, Laboratory Medicine Building, Queen Elizabeth University Hospital Greater Glasgow & Clyde NHS, Glasgow, UK. [8]Department of Surgery, Oncology and Gastroenterology, University of Padua, Padua, Italy. [9]ARC-Net Research center, University of Verona, Verona, Italy. [10]Section of Oncology, Department of Engineering for Innovation Medicine, University of Verona Hospital Trust, Verona, Italy. [11]NIHR Manchester Biomedical Research Centre, Manchester, UK. [12]These authors contributed equally: Antonio Pea, Xiaotong He. [13]These authors jointly supervised this work: David C. Wedge, David K. Chang. ✉e-mail: antonio.pea@univr.it; David.Wedge@manchester.ac.uk; David.Chang@glasgow.ac.uk

