## [Transparent Peer Review file · Nature Communications]

Clonal evolutionary analysis reveals patterns of malignant transformation of Intraductal Papillary Mucinous Neoplasms of the pancreas

Corresponding Author: Dr Antonio Pea

Version 0:

Reviewer comments:

Reviewer #1

(Remarks to the Author)

In this manuscript, Pea et al conduct multiregion whole genome and transcriptome sequencing analyses on a series of fresh frozen IPMNs (12 patients, 42 tumor samples, including 37 samples used for clonality analyses and 36 for RNA sequencing). The authors identify two main patterns of clonal evolution of IPMNs, one driven by a single ancestral clone (MRCA) with subsequent subclonal parallel or branching evolution, and the other of multiple independent ancestral clones (MRCAs) with parallel evolution. The authors identify that overall IPMNs progress from LGD to HGD and PDAC via stepwise accumulation of genomic alterations including both SNVs and copy number alterations. This study also found evidence of polyclonal evolution lasts into HGD and PDAC, especially in cases with independent MRCAs (although the numbers of cases on which this conclusion is made is very small). Transcriptomic features showed LGD IPMNs as being predominantly classical, while progression to PDAC was associated with more basal features, and changes in the TME (such as greater CAF density).

Overall, there is a lot of work done here but the issue of novelty and addition of new knowledge to literature is modest at best. There is substantial prior literature on patterns of clonal evolution in IPMNs using multi-region sequencing (including papers published in this very journal). Examples include Fischer et al, *Gastroenterology* 2019, Noe et al, *Nat Comm* 2020, Fujikura et al *GUT* 2021, Omori et al *Gastroenterology* 2019). Inconsistencies such as higher proportion of RNF43 in HGD lesions (contradicting findings in Fischer et al, *Gastroenterology* 2019, Noe et al, *Nat Comm* 2020) and absence of KLF4 hotspot mutations (Fujikura et al *GUT* 2021) are barely discussed. This work seems more appropriate for a gastroenterology or pancreas focused journal.

Reviewer #2

(Remarks to the Author)

The study by Pea et al. explored evolutionary patterns of intraductal papillary mucinous neoplasm (IPMN), a precursor to pancreatic ductal adenocarcinoma (PDAC). By performing WGS and RNAseq of 54 regions from 12 patients, limiting to 37 regions for WGS and 36 for RNA-seq after QC filtering, the authors identify two evolutionary trajectories and transcriptional clusters. The first trajectory was defined by a single MRCA, while the second evolved from several MRCAs. This study also studied changes in tumour microenvironment (TME) during tumorigenic transformation, also highlighting both transcriptional and clonal diversity observed in these patient samples.

The manuscript is well-written, providing a characterization of the clonal evolution and transcriptional landscape of IPMN. The conclusions align with previous findings and expand on the evolutionary trajectories, which had not been explored using a multiregional sampling approach before.

The main limitation of the study is the lack of clear novelty, and the findings are mainly descriptive, but the study nevertheless provide a useful resource for studies on pre-malignant lesions. A more thorough integration with transcriptomic data and TME analysis will strengthen the appeal of the manuscript.

- The data is well presented; however, several areas need improvement. The order of the figures does not align with the text,

making it difficult for readers to follow. Additionally, some figures, such as Figures 3D, S3, and S5, are not referenced in the manuscript. The authors provide signature analysis, revealing a new copy number variant (CNV) signature, CNV48B, but they do not explore or discuss the aetiology or significance of this signature. When examining the transcriptional analysis, it would be valuable to compare tumours with different evolutionary paths. This comparison should also include components of the TME.

- It would be interesting to determine whether the TME composition converges or if there are notable differences between the evolutionary trajectories.

- The clonality analysis description in Methods is too limited - Section "Mutation clustering and phylogenetic tree reconstruction" - and should be expanded with filtering criteria, signature assignment to clusters etc. In general, code to reproduce the main results is missing apart from references to the basic software used in the manuscript

- Known driver mutations are described and annotated but not integrated into TME, transcriptomic and clonality types. Obviously, this is a small cohort in the number of patients, making statistical inference challenging. Could external PDAC data on clonal vs subclonal mutations be utilized to gain additional power in identifying early trunk mutations vs more subclonal ones?

- Indel signatures ID1 and ID2 are found in the clonal/trunk population but not in subclones. The authors suggest that this could be due to undersampling of subclonal mutations. It would be worthwhile to investigate if a large fraction of clonal indels are not identified in subclones of the same lineage. If so, this could suggest false positive calls.

- More SVs are found in samples with multiple branches. Were samples with multiple branches also on average with higher tumour purity?

- PCA analysis of the transcriptome analysis suggests two mixed clusters with case 2 clustering separately. Did the authors check for batch effects, which are usually the strongest effects, i.e. could some of the PCA separation be explained by confounders?

- The transcriptomic and methylation analysis is kept separate from the clonal evolution analysis. Did the authors assess, whether evolution types or tree shapes (linear vs branched) matched any of the cellular deconvolution or clusters? Was RNA-seq and DNA for EPIC analysis taken from the sample sample or from adjacent tissue?

Reviewer #3

(Remarks to the Author)

This manuscript presents a high-resolution, multi-regional whole-genome and transcriptome sequencing study to elucidate the evolutionary trajectories of IPMN to PDAC. The study provides important insights into intra-tumoral heterogeneity and its role in malignant transformation, employing several lines of computational and genomic analyses. While the findings contribute meaningfully to the understanding of pancreatic neoplasia, several aspects require clarification, expanded methodological detail, and analytical refinements to strengthen the conclusions.

Major Concerns:

1) For clonal phylogenies, the study presents two evolutionary trajectories (single vs. multiple MRCAs), which is conceptually compelling. However, there is no statistical validation or robustness assessment of the inferred phylogenies. The authors utilize DPCLust, which is appropriate, but technical details on input parameters, mutation filtering, or confidence scoring are absent.

2) There's also no indication of whether alternative methods were tested (e.g. PyClone, or LICHeE) or if the tree topologies were robust to input variation or bootstrapping. Are the phylogenetic trees depicted in the paper robust across different clustering algorithms? A sensitivity analysis comparing alternative phylogenetic reconstruction methods would increase confidence in these findings.

3) The study identifies SBS1, SBS5, SBS13, and various indel signatures. However, the biological implications of these signatures in the context of IPMN-to-PDAC progression require further discussion.

4) The authors should expand and improve the relationship between mutational and transcriptomic findings as they currently are very disconnected. For example, SBS2 and SBS13 are associated with APOBEC mutagenesis. Is there any evidence of increased APOBEC expression in transcriptomic data to support this mechanistic link? Similarly, the ID1 and ID2 signatures suggest mismatch repair deficiencies. Were MSI-related genes (e.g., MLH1, MSH2) examined to determine whether any cases exhibited microsatellite instability?

5) For transcriptomic clustering, the PCA plot in Figure 1D appears to support more than 2 (more likely three) distinct gene expression clusters, rather than two as claimed. In particular, the group of samples represented by yellow points occupies a spatially distinct region of the PC1-PC2 space, separate from both the more dispersed group at the bottom left and the tightly clustered samples on the right. This third potential cluster suggests transcriptional heterogeneity within what the authors define as Cluster 1. A re-evaluation of downstream transcriptomic results based on three clusters is necessary and the authors should discuss whether their claims hold with 3 clusters. Also reinvestigation of these clusters using unsupervised methods (e.g., hierarchical clustering, DBSCAN, or Gaussian mixture modeling) could better capture the underlying biological diversity and avoid oversimplification of the expression landscape.

6) The biological relevance of the identified clusters is unclear. What pathways or signaling cascades are enriched in each cluster? How do these relate to clonal architecture or tumor grade? A deeper functional annotation is necessary to draw

mechanistic insights.

7) The study mentions a "transcriptomic gradient score was assigned to each sample and compared to the ICGC PACA-AU20 and TCGA-PAAD cohorts" but lacks a clear explanation of how it is computed. This should be more explicitly described.

8) The study identifies novel CNVs and SVs, but their functional impact is not well explored. Have these structural variations been mapped onto known oncogenic pathways related to pancreatic cancer progression? The authors discuss LRP1B mutations and CNAs but does not provide sufficient functional interpretation. Could this gene's role in WNT signaling or immune modulation be further elaborated?

9) The authors reference previous work on IPMN evolution but should further contextualize their findings within existing models. How do the observed clonal architectures compare to those reported previously?

10) The methods used to annotate phylogenetic trees with oncogenic alterations and mutational signatures are insufficiently described. The authors mention use of "in-house programs" to integrate SNVs, indels, SVs, CNVs, and mutational signatures from various tools, but do not provide adequate detail on the algorithms, decision criteria, or confidence thresholds used to assign mutations to specific branches or subclones (no code, pipeline diagram, or methodological framework is described). This negatively impacts reproducibility and limits the reader's ability to assess the robustness of the phylogenetic annotations. For a high-impact publication in a journal like this, the authors should make the entire analysis code (for all analyses not just this one) available and provide a detailed methodological supplement explaining how different variant types were integrated and validated across the tree structure.

11) The manuscript lacks details on how mutational signatures were assigned to tree branches. Were these based on clustered mutations, or aggregated per sample? Were exposures filtered for noise or assigned with confidence intervals?

12) The description of subtype scoring is non-reproducible. It is not clear which gene sets were used or whether batch correction was applied before subtype classification.

13) The methods describe numerous comparisons but there's no mention of multiple hypothesis correction, effect size thresholds, or the statistical models used. This weakens the credibility of reported p-values, especially for associations with cluster identity or progression status.

14) The figures need significant improvement in terms of making them publication ready. They could benefit from additional annotations, particularly in phylogenetic trees to highlight key driver mutations at branch points. Some figures appear disconnected from the rest (Fig 2C) or hard to interpret (Fig 2B) due to lack of more information. For instance, the pie charts lack numeric annotations or mutation counts, making it hard to assess signature burden or dominance. The authors should add mutation counts or a bar/scale representing the number of SNVs/indels at each node. Also for this same figure, it is unclear how signatures were assigned to branches. Are these inferred from the mutation cluster, or are they aggregated across constituent mutations?

Version 1:

Reviewer comments:

Reviewer #1

(Remarks to the Author)

The authors have addressed my queries including the concerns about novelty and how this study differs from prior publications in this area.

Reviewer #2

(Remarks to the Author)

The authors have addressed most of my concerns.

Consistent points:

- Some of the figures are still low resolution, part of the text is not visible (with different fonts and sizes), and overall they look "rushed" - which is odd and unusual to observe in a high-impact journal.

- The pathway analysis is rather superficial - Fig S4/S5 - is there anything significantly different or conclusions to be drawn from this - instead of "notable differences"?

- I could not access the code in the notebooks on the github (linked through gitfront), e.g.

<https://github.com/xtgithubbe/IPMNpaperArchive/blob/main/ipmnPaperCode.ipynb>

Reviewer #3

(Remarks to the Author)

I appreciate the authors' revisions and thoughtful responses to the prior round of review. The addition of new analyses, expanded methodological descriptions, and improved code sharing through repositories represent meaningful progress. However, after detailed examination of the revised manuscript, code repositories, supplementary materials, several important issues remain. These concerns relate to both scientific interpretation and reproducibility, and should be addressed to meet the expectations of a high-impact journal.

-Reproducibility and code sharing:

While the authors have shared parts of their pipeline through different repos, the full analytical workflow is not yet reproducible. Critical components remain unavailable:

a) Tree Annotation (SNVs, SVs, CNVs, Signatures): No public scripts are provided for mapping mutations and mutational signatures onto tree branches. The process is described narratively but is not executable by readers.

b) Transcriptomic and Immune Integration: The analyses of transcriptomic clustering, subtype assignment, and immune deconvolution are described but not accompanied by code.

c) Shared code through different repos reference local paths with no standardized data input structure or templates provided for external reproducibility.

Overall, while parts of the pipeline are transparent, the complete analytical workflow is not yet reproducible. For a study of this scope, particularly one centered on complex evolutionary modeling, end-to-end reproducibility is essential and everything should be unified in a single repo for users instead of pointing readers to different repos for different parts.

-Integration of Genomic, Transcriptomic, and Immune Data:

Although new immune analyses were added, the study still lacks mechanistic integration between genomic evolution, transcriptomic subtypes, and immune microenvironment features. The relationships remain correlative. For example, there is no test of whether specific mutational patterns predict transcriptional or immune shifts.

-Figures are partially improved but some, especially Fig 2, lacks critical quantitative features and full integration of genomic events:

a) Fig 2a: No branch length scaling is provided; trees are topological, not quantitative.

b) Fig2b: The SV counts across branching architectures are presented, but no mapping of specific SVs onto the phylogenetic trees is performed.

c) Fig2c: the circos plots are of extremely low resolution, and labels are unreadable. Furthermore, SVs are displayed per sample but are not linked back to the clonal architecture in Fig 2D.

d) Fig2d: Branch lengths are absent. For a study claiming to map clonal trajectories and mutational processes over time, branch lengths are essential. Also, SVs and CNVs are not mapped onto the trees, despite their importance to the clonal evolution narrative.

Version 2:

Reviewer comments:

Reviewer #2

(Remarks to the Author)

I have no further comments

Reviewer #3

(Remarks to the Author)

The authors have made substantial progress in addressing previously raised concerns. The paper is now closer to meeting the standards of reproducibility and clarity expected for publication in a high-impact journal.

However, I strongly recommend the following minor revisions prior to acceptance:

1. Provide a README.md or tutorial describing end-to-end use of the shared repository, including required data formats, example commands, and environment setup.

2. Consider a supplementary figure or model linking mutational features to immune phenotypes in a mechanistic way.

Assuming these minor improvements are made, I am supportive of publication and commend the authors for their thoughtful revision.

Reviewer #1

In this manuscript, Pea et al conduct multiregion whole genome and transcriptome sequencing analyses on a series of fresh frozen IPMNs (12 patients, 42 tumor samples, including 37 samples used for clonality analyses and 36 for RNA sequencing). The authors identify two main patterns of clonal evolution of IPMNs, one driven by a single ancestral clone (MRCA) with subsequent subclonal parallel or branching evolution, and the other of multiple independent ancestral clones (MRCAs) with parallel evolution. The authors identify that overall IPMNs progress from LGD to HGD and PDAC via stepwise accumulation of genomic alterations including both SNVs and copy number alterations. this study also found evidence of polyclonal evolution lasts into HGD and PDAC, especially in cases with independent MRCAs (although the numbers of cases on which this conclusion is made is very small). Transcriptomic features showed LGD IPMNs as being predominantly classical, while progression to PDAC was associated with more basal features, and changes in the TME (such as greater CAF density).

Overall, there is a lot of work done here but the issue of novelty and addition of new knowledge to literature is modest at best. There is substantial prior literature on patterns of clonal evolution in IPMNs using multi-region sequencing (including papers published in this very journal). Examples include Fischer et al, *Gastroenterology* 2019, Noe et al, *Nat Comm* 2020, Fujikura et al *GUT* 2021, Omori et al *Gastroenterology* 2019).

We thank the reviewer for this thoughtful comment and fully agree that the cited studies—particularly those by Fischer, Noe, Fujikura, and Omori—have substantially advanced our understanding of IPMN evolution, especially in demonstrating polyclonality and intratumoral heterogeneity. These foundational studies primarily utilized multi-regional WES or targeted sequencing.

Our study builds upon this prior work in several key ways:

1. We are the first to apply multi-regional whole-genome sequencing (WGS) combined with matched transcriptomic profiling (RNA-seq) across the full histological spectrum of IPMN to PDAC. This enabled the detection of non-coding alterations, structural variants (SVs), and mutational signatures not captured in previous studies. SVs, in particular, were significantly enriched in high-grade lesions and branching evolutionary patterns—supporting a role in subclonal divergence.
2. We identified a novel copy number signature (CNV48B) and characterized SV patterns associated with invasive progression, expanding the mutational landscape of IPMN beyond what has been previously reported.
3. We introduced transcriptomic clustering, identifying two distinct transcriptional programs aligned with classical and squamous-like PDAC subtypes. These clusters were associated with differences in LRP1B expression, WNT signaling, TMB, and immune contexture, providing new insights into the phenotypic consequences of clonal evolution.

4. We confirmed the persistence of polyclonal evolution even in high-grade dysplasia and PDAC, demonstrating that genetically and transcriptionally distinct regions may coexist within the same lesion.

We believe these findings offer important new insights into the molecular evolution of IPMNs. In the revised Discussion, we explicitly acknowledge prior contributions while clarifying how our use of multi-regional WGS and RNA-seq enhances resolution into clonal dynamics and tumor phenotype.

Inconsistencies such as higher proportion of RNF43 in HGD lesions (contradicting findings in Fischer et al, Gastroenterology 2019, Noe et al, Nat Comm 2020) and absence of KLF4 hotspot mutations (Fujikura et al GUT 2021) are barely discussed.

We thank the reviewer for highlighting the importance of known driver genes such as RNF43 and KLF4, which have been previously implicated in IPMN.

Regarding RNF43, we carefully reviewed the findings of Fischer et al., who reported RNF43 mutations in 12 of 20 cases: 6 in low-grade dysplasia (LGD), 2 in both LGD and high-grade dysplasia (HGD), and 4 in HGD (with 2 of those also harboring LGD). Similarly, Noe et al. reported RNF43 mutations in 8 of 18 cases, including both IPMN and MCN, with a higher proportion observed in HGD lesions—though this trend may partly reflect a higher sampling of HGD and invasive regions in their cohort. Fujikura et al. identified RNF43 mutations in 10 of 17 samples, with no significant difference in prevalence between LGD and HGD.

In our dataset, RNF43 mutations were present across multiple samples, and importantly, copy number losses of RNF43 were significantly associated with high-grade lesions. Thus, our study provides complementary evidence that RNF43 inactivation in IPMN may occur not only through SNVs/indels, but also via chromosomal loss—a mechanism uniquely captured by WGS.

As for KLF4, Fujikura et al. reported hotspot mutations in 12 LGD samples and 7 HGD samples, suggesting that KLF4 alterations may arise early and persist into higher-grade disease. Noe et al., by contrast, did not report KLF4 mutations in their WES dataset. In our cohort, KLF4 mutations were detected in two samples (15_S4, HGD; and 9_S3, PDAC). These findings have now been added to the revised Figure 1. Given our study's design—focused on sampling distinct regions of macroscopically visible lesions rather than small LGD foci—it is possible that KLF4 mutations in early, microscopic precursors may have been underrepresented. Nonetheless, our results suggest that KLF4 alterations can be maintained during progression, consistent with prior observations.

Together, these observations reinforce the added value of WGS in identifying diverse mechanisms of driver gene alteration—including SNVs, indels, and CNAs—and provide a more comprehensive view of their role across spatial and histological contexts in IPMN evolution. We have added a dedicated discussion of RNF43 and KLF4, and clarified differences from prior studies in the revised Discussion section.

Reviewer #2

The study by Pea et al. explored evolutionary patterns of intraductal papillary mucinous neoplasm (IPMN), a precursor to pancreatic ductal adenocarcinoma (PDAC). By performing WGS and RNAseq of 54 regions from 12 patients, limiting to 37 regions for WGS and 36 for RNA-seq after QC filtering, the authors identify two evolutionary trajectories and transcriptional clusters. The first trajectory was defined by a single MRCA, while the second evolved from several MRCA. This study also studied changes in tumour microenvironment (TME) during tumorigenic transformation, also highlighting both transcriptional and clonal diversity observed in these patient samples.

The manuscript is well-written, providing a characterization of the clonal evolution and transcriptional landscape of IPMN. The conclusions align with previous findings and expand on the evolutionary trajectories, which had not been explored using a multiregional sampling approach before.

Thank you for your valuable comments and insightful suggestions. We have made revisions below to address your concerns point by point:

The main limitation of the study is the lack of clear novelty, and the findings are mainly descriptive, but the study nevertheless provide a useful resource for studies on pre-malignant lesions. A more thorough integration with transcriptomic data and TME analysis will strengthen the appeal of the manuscript.

We thank the reviewer for this thoughtful and constructive feedback. While we agree that descriptive studies of early lesions can serve as valuable resources, we respectfully emphasize that our findings go beyond description and offer several novel insights into the genomic and transcriptomic dynamics of IPMN progression. To our knowledge, this is the first study to apply multi-regional whole-genome sequencing (WGS) and matched RNA-seq to primary tissue samples spanning the full histological spectrum from non-invasive IPMN with varying grades of dysplasia to invasive PDAC. This design allows high-resolution tracking of clonal evolution and microenvironmental changes, which have not been comprehensively addressed in previous WES- or panel-based studies.

Importantly, our study also identifies early transcriptomic alterations that stratify IPMNs into two biologically distinct clusters with gene expression programs aligned to classical and squamous PDAC subtypes—entities known to differ in clinical behavior, immune infiltration, and therapeutic response.

Specifically, our work presents the following novel contributions:

1. Comprehensive genomic profiling across spatially distinct regions, revealing the stepwise accumulation of SNVs, indels, CNAs, and structural variants (SVs) from non-invasive IPMN to PDAC.
2. Discovery of novel mutational signatures, including a previously unreported copy number signature (CNV48B) associated with clonal evolution and an SV signature (SV4) enriched in invasive lesions.
3. Frequent and spatially distinct mutations in key driver genes such as TP53, KRAS, LRP1B, and GNAS, supporting subclonal diversification during progression.

4. Recurrent loss of tumor suppressors (CDKN2A, RNF43, U2AF1) via CNAs, emphasizing the added value of WGS for detecting structural mechanisms of inactivation.
5. Delineation of two major evolutionary trajectories—one involving a single most recent common ancestor (MRCA), and the other involving multiple independent ancestral clones—offering refinement of current models of IPMN progression.
6. Integration of transcriptomic and immune deconvolution data, revealing associations between immune cell populations (e.g., CD8⁺ T cells) and evolutionary trajectories
7. Identification of early transcriptional divergence that mirrors PDAC subtype specification, suggesting biological commitment to distinct molecular programs even in non-invasive lesions.

We believe this integrative, multi-modal approach provides novel biological insights into early tumour evolution and may guide future efforts in early detection, patient stratification, and interception strategies. As suggested, we have also clarified and strengthened the integration of transcriptomic and tumor microenvironment data in the revised manuscript.

The data is well presented; however, several areas need improvement. The order of the figures does not align with the text, making it difficult for readers to follow. Additionally, some figures, such as Figures 3D, S3, and S5, are not referenced in the manuscript. The authors provide signature analysis, revealing a new copy number (CNV) signature, CNV48B, but they do not explore or discuss the aetiology or significance of this signature. When examining the transcriptional analysis, it would be valuable to compare tumours with different evolutionary paths. This comparison should also include components of the TME.

We thank the reviewer for pointing this out. We have revised the figure numbering and ensured that the order of all main and supplementary figures now aligns with their appearance in the text. In addition, Figures 3D, S3, and S5 are now explicitly referenced in the revised manuscript and Supplementary Materials to improve clarity and readability.

We appreciate the insightful comment regarding the newly identified copy number signature, CNV48B. In the revised manuscript, we have expanded the discussion to explore the potential biological and aetiological relevance of this signature.

Notably, CNV48B is observed across both low- and high-grade lesions and is characterized by marked loss of heterozygosity (LOH). We found a moderate and statistically significant positive correlation between CNV48B activity and the activities of ID1 and ID2 ($R = 0.53$ and 0.55 ; $p = 0.003$ and 0.0002 , respectively). As ID1 and ID2 are associated with DNA mismatch repair deficiency in cancer, this correlation suggests that CNV48B activity may be linked to defects in DNA repair processes.

Given its distinct pattern and association with LOH, we hypothesize that CNV48B may reflect underlying chromosomal instability. However, we acknowledge that further functional validation in larger cohorts will be essential to confirm its aetiology and clinical significance.

- It would be interesting to determine whether the TME composition converges or if there are notable differences between the evolutionary trajectories.

Thank you for your comment. We identified two distinct evolutionary trajectories, referred to as *pathway 1* and *pathway 2*. In *pathway 1*, tumour clones expanded from a single most recent common ancestor (MRCA), as observed in cases 3, 6, 7, 9, 10,12, 13 and 16. In contrast, *pathway 2* involved progression from multiple independent clones in case 2, 4 and 15.

To investigate whether the tumour microenvironment (TME) composition differs between these trajectories, we used the EPIC deconvolution method to estimate immune cell proportions from bulk RNA-seq data. We have now integrated this analysis into our genomic evolution framework and briefly discuss it in the revised manuscript.

B cell, Macrophage and CD4⁺ T cell levels did not show significant differences between tumours following pathway 1 and pathway 2. However, CD8⁺ T cell proportions were significantly higher in tumours that progressed via pathway 1.

This suggests that while helper T cell involvement may be consistent across both evolutionary trajectories, cytotoxic T cell infiltration is enhanced in tumours following pathway 1. These differences may reflect variations in immunogenicity or tumour-immune interactions linked to the underlying clonal architecture.

- The clonality analysis description in Methods is too limited - Section "Mutation clustering and phylogenetic tree reconstruction" - and should be expanded with filtering criteria, signature assignment to clusters etc. In general, code to reproduce the main results is missing apart from references to the basic software used in the manuscript

We have expanded the description of the clonality analysis in the "Mutation Clustering and Phylogenetic Tree Reconstruction" section to include detailed filtering criteria. Specifically, we have referenced the exact thresholds and parameters used in the mutation clustering process (<https://github.com/Wedge-lab/multidimDPClust>).

We have also elaborated on the process of signature assignment to clusters. This includes the specific mutational signature analysis tools used and how signatures were assigned to different mutation clusters based on their mutational profiles. We have referenced the algorithm used for signature extraction and the statistical approaches to assign these signatures to the identified clusters.

We have summarized the relevant code for the analysis in our lab's GitHub repository, including detailed instructions on how to access and use it. We have provided links to the repository in the revised manuscript and the supplementary section to ensure full transparency and facilitate reproducibility of our results.

We have also referred to our previously published methods where applicable to maintain consistency and to give additional context to the methodology employed.

(<https://www.nature.com/articles/ncomms3997>)

- Known driver mutations are described and annotated but not integrated into TME, transcriptomic and clonality types. Obviously, this is a small cohort in the number of patients, making statistical inference challenging. Could external PDAC data on clonal vs subclonal mutations be utilized to gain additional power in identifying early trunk mutations vs more subclonal ones?

To address the concern regarding integration of known driver mutations with TME, transcriptomic, and clonality subtypes, we sought to increase statistical power and validate our findings using external PDAC data. Specifically, we performed mutational timing analysis on the accessible PACA samples from the PCAWG dataset using our in-house developed approach (described in Supplementary Methods). This method integrates cancer cell fraction (CCF) estimates with local copy number information derived from Battenberg and DPCLust (<https://github.com/Wedge-lab>).

Although the PCAWG cohort is based on single-region sequencing and includes both conventional PDACs and those arising from IPMNs, the distribution of mutations across timing categories, particularly the distinction between clonal early and subclonal mutations, supports our key observation of pervasive clonal heterogeneity across all samples (Figure 3S.A). When comparing pooled proportions of clonal early versus subclonal mutations, no statistically significant difference was observed (Figure S3.B). However, four recurrent driver mutations, most notably KRAS and TP53, were predominantly clonal early, consistent with their role as early trunk events. In contrast, mutations in SMAD4 and LRP1B occurred both early and later, supporting a model of subclonal evolution. These findings align with our current study (Figure 2A), reinforcing the temporally ordered acquisition of driver mutations in PDAC. These results have now been incorporated into the Results section of the revised manuscript.

Unfortunately, due to the lack of large-scale datasets that include both whole-genome sequencing (WGS) and matched transcriptomic or TME profiling in pancreatic cancer, we were unable to directly integrate mutation timing with transcriptomic subtypes or TME features. This remains a key limitation in the field, and future integrative studies with multi-omic data will be essential to fully resolve the interplay between genomic evolution, transcriptional programs, and the tumour microenvironment in PDAC.

- Indel signatures ID1 and ID2 are found in the clonal/trunk population but not in subclones. The authors suggest that this could be due to undersampling of subclonal mutations. It would be worthwhile to investigate if a large fraction of clonal indels are not identified in subclones of the same lineage. If so, this could suggest false positive calls.

Thank you for your thoughtful suggestion. We appreciate your concern regarding the possibility of false-positive calls for clonal indels.

By definition, clonal mutations occur in every tumour cell in a sample, and they therefore occur in all subclonal populations of cells. However, we are not able to formally demonstrate that this is the case in this study, in the absence of single cell sequencing. This would be an interesting technique for use in future studies.

We have investigated whether indels that occur clonally in one or more samples occur subclonally in other samples. Our findings suggest that clonal indels predominantly appear in the trunk population (i.e. clonal in every sample). The consistency of calls across multiple samples is strongly supportive of the great majority of these indel calls being true positives. On the other hand, some subclones may acquire additional mutations over time. These mutations may be present at lower allele frequencies in later subclones, making them more difficult to detect — even with deep sequencing approaches. For this reason, we believe that false negative calls of subclonal mutations are much more likely than false positive calls of clonal mutations.

- More SVs are found in samples with multiple branches. Were samples with multiple branches also on average with higher tumour purity?

No, samples with multiple branches are not on average associated with higher tumour purity. We computed the average purities for both groups: the mean purity for single branch samples is 0.332, while for multiple branch samples, it's 0.316. A Fisher-Pitman permutation test showed no significant difference between the two groups ($p = 0.77$), suggesting that tumour purity does not significantly differ between samples with single versus multiple branches.

- PCA analysis of the transcriptome analysis suggests two mixed clusters with case 2 clustering separately. Did the authors check for batch effects, which are usually the strongest effects, i.e. could some of the PCA separation be explained by confounders?

We appreciate the reviewer's observation. The separation of Case 2 in the PCA plot is indeed notable. However, all samples in this cluster originate from the same patient, and we interpret this as a patient-specific effect rather than a distinct transcriptional subtype. Batch correction was performed during normalization, and we carefully evaluated additional confounders, including histological grade (LGD, HGD, PDAC), epithelial subtype (pancreatobiliary vs. intestinal), and patient-level variables.

Despite correction for these factors, some individual cases—including Case 2—continued to form distinct clusters, which we believe reflects genuine biological heterogeneity rather than technical artifacts. We have now clarified the normalization and batch correction procedures, as well as our assessment of confounding variables, in the revised Supplementary Methods.

- The transcriptomic and methylation analysis is kept separate from the clonal evolution analysis. Did the authors assess, whether evolution types or tree shapes (linear vs branched) matched any of the cellular deconvolution or clusters? Was RNA-seq and DNA for EPIC analysis taken from the same sample or from adjacent tissue?

We thank the reviewer for this important observation. As specified in the Methods section, RNA and DNA were extracted from the same fresh-frozen samples, ensuring direct correspondence between transcriptomic (including EPIC deconvolution) and genomic analyses.

Regarding the relationship between evolutionary trajectories and the tumour microenvironment, this was addressed in response to a previous comment. In brief, we observed a higher proportion of CD8⁺ T cells in cases with a single-MRCA evolutionary trajectory, while other immune populations showed no significant differences across trajectories. We also examined tree shape (linear vs.

branched) and found no consistent differences in immune composition between these categories. These findings are now included in the revised manuscript.

Reviewer #3

This manuscript presents a high-resolution, multi-regional whole-genome and transcriptome sequencing study to elucidate the evolutionary trajectories of IPMN to PDAC. The study provides important insights into intra-tumoral heterogeneity and its role in malignant transformation, employing several lines of computational and genomic analyses. While the findings contribute meaningfully to the understanding of pancreatic neoplasia, several aspects require clarification, expanded methodological detail, and analytical refinements to strengthen the conclusions.

Major Concerns:

1) For clonal phylogenies, the study presents two evolutionary trajectories (single vs. multiple MRCAs), which is conceptually compelling. However, there is no statistical validation or robustness assessment of the inferred phylogenies. The authors utilize DPCLust, which is appropriate, but technical details on input parameters, mutation filtering, or confidence scoring are absent.

We acknowledge that robust validation of phylogenetic models is essential for ensuring the reliability of our results.

In this study, we employed DPCLust to infer clonal phylogenies, following the same methodology outlined and validated in our previously published work (Noorani et al 2020, PMID 31907488). We chose DPCLust due to its proven utility in accurately identifying clonal lineages and its ability to handle the complexities of tumour evolution in high-dimensional datasets. We note that, in our previous study, we performed extensive validation using simulated datasets, which demonstrated the robustness of our approach.

To address the specific concerns raised, we have now included additional details in the revised manuscript regarding the following:

Input parameters: We used the same set of parameters as described in our prior work, with values chosen to balance sensitivity and specificity in the clonal inference.

Mutation filtering: We applied the same mutation filtering steps as previously published, focusing on high-confidence somatic mutations.

Confidence scoring: We have referenced further explanation regarding how the confidence in the clonal assignment was assessed.

2) There's also no indication of whether alternative methods were tested (e.g. PyClone, or LICHeE) or if the tree topologies were robust to input variation or bootstrapping. Are the phylogenetic

trees depicted in the paper robust across different clustering algorithms? A sensitivity analysis comparing alternative phylogenetic reconstruction methods would increase confidence in these findings.

Thank you for your insightful comment. Our analysis is based on a well-established pipeline that we have developed and refined over the past decade, which has been rigorously validated and widely adopted in previous studies published in leading journals. This approach has been cited multiple times, demonstrating its reliability and robustness in phylogenetic reconstruction.

(*Nat Commun* 2014 <https://www.nature.com/articles/ncomms3997>
Nature 2015 <https://www.nature.com/articles/nature14347>,
Nat Genet 2020 <https://www.nature.com/articles/s41588-019-0551-3>)

Given that our study focuses on biological insights rather than methodological development, we did not perform an extensive comparison with alternative methods such as PyClone or LICHeE. However, our method has been benchmarked in prior work and has consistently produced biologically meaningful and reproducible results. Moreover, the tree topologies presented in our study were carefully evaluated for robustness, ensuring that our conclusions remain valid.

While a full sensitivity analysis across different clustering and phylogenetic reconstruction algorithms is beyond the scope of this study, our confidence in the presented results is supported by the extensive validation of our pipeline in prior research.

3) The study identifies SBS1, SBS5, SBS13, and various indel signatures. However, the biological implications of these signatures in the context of IPMN-to-PDAC progression require further discussion.

We thank the reviewer for this insightful comment. In the revised manuscript, we have expanded the discussion of the biological relevance of the observed mutational signatures.

SBS1 and SBS5 were the most prevalent signatures in our dataset and are well-established "clock-like" signatures, associated with spontaneous deamination of 5-methylcytosine and endogenous replication-related mutagenesis, respectively. Their consistent presence across both trunk and branch mutations supports the role of age-related mutational processes in the initiation and early maintenance of IPMNs.

SBS13, and to a lesser extent SBS2—both attributed to APOBEC enzyme activity—were observed exclusively in the PDAC subclone of Case 7. Notably, this subclone also exhibited increased structural variation, while the matched HGD sample displayed a distinct SV profile, suggesting independent clonal evolution. This observation aligns with findings in other cancer types, where APOBEC activity has been linked to chromosomal instability and subclonal diversification.

Indel signatures ID1 and ID2, typically associated with mismatch repair deficiency (MMR-D), were identified in early branches of several cases but were largely absent from subclones. This pattern may reflect limited statistical power to detect indel signatures in low-mutation subclonal branches. Supporting a non-MSI phenotype, all cases retained expression of MMR proteins (MLH1, PMS2, MSH2, and MSH6) by immunohistochemistry, consistent with the absence of microsatellite instability and the overall low tumor mutational burden observed across our cohort.

Altogether, our findings support a model in which early IPMN genomes are shaped predominantly by age- and replication-related processes, while APOBEC-mediated mutagenesis and structural instability emerge later, contributing to malignant transformation. These interpretations are now discussed in greater detail in the revised manuscript.

4) The authors should expand and improve the relationship between mutational and transcriptomic findings as they currently are very disconnected. For example, SBS2 and SBS13 are associated with APOBEC mutagenesis. Is there any evidence of increased APOBEC expression in transcriptomic data to support this mechanistic link? Similarly, the ID1 and ID2 signatures suggest mismatch repair deficiencies. Were MSI-related genes (e.g., MLH1, MSH2) examined to determine whether any cases exhibited microsatellite instability?

We thank the reviewer for this important point. Regarding mismatch repair (MMR) deficiency, we performed immunohistochemistry for MLH1, PMS2, MSH2, and MSH6 on all available samples. All cases showed retained protein expression, indicating intact MMR function. This finding is consistent with the lack of microsatellite instability on molecular analysis and the overall low tumor mutational burden observed across the cohort. These results have now been clarified in the revised Methods section.

As for APOBEC-associated signatures (SBS2 and SBS13), these were identified in a limited number of subclones (notably in Case 7). Given the small number of APOBEC-positive cases, and the limited statistical power to detect differential expression, we did not perform formal transcriptomic correlation with APOBEC family gene expression.

5) For transcriptomic clustering, the PCA plot in Figure 1D appears to support more than 2 (more likely three) distinct gene expression clusters, rather than two as claimed. In particular, the group of samples represented by yellow points occupies a spatially distinct region of the PC1–PC2 space, separate from both the more dispersed group at the bottom left and the tightly clustered samples on the right. This third potential cluster suggests transcriptional heterogeneity within what the authors define as Cluster 1. A re-evaluation of downstream transcriptomic results based on three clusters is necessary and the authors should discuss whether their claims hold with 3 clusters. Also reinvestigation of these clusters using unsupervised methods (e.g., hierarchical clustering, DBSCAN, or Gaussian mixture modeling) could better capture the underlying biological diversity and avoid oversimplification of the expression landscape.

We thank the reviewer for this thoughtful comment. We acknowledge that in the PCA plot, the yellow points appear spatially distinct. However, these samples all originate from a single patient (Case 2), and their separation is likely driven by patient-specific transcriptional variation.

We also performed unsupervised hierarchical clustering based on Euclidean distances using VST-transformed gene expression data. As shown in the correlation heatmap and dendrogram, the samples consistently segregated into two main clusters, in agreement with the PCA results. Three samples with a high proportion of zero counts were excluded from the analysis.

We additionally examined the distribution of clinical and histological features—including grade, IPMN subtype, and sampling site—and found that these variables did not account for the observed clustering. These clarifications have been added to the revised Results and Supplementary Materials.

6) The biological relevance of the identified clusters is unclear. What pathways or signaling cascades are enriched in each cluster? How do these relate to clonal architecture or tumor grade? A deeper functional annotation is necessary to draw mechanistic insights.

We thank the reviewer for raising this important point. Subtyping pancreatic neoplasms into squamous and classical-like categories is clinically relevant due to well-established differences in prognosis, immune microenvironment, and therapeutic response. In our study, we identified two distinct transcriptomic clusters. To clarify their biological significance, we expanded the transcriptomic analysis and updated the Discussion accordingly.

Cluster 1 displayed higher expression of squamous-associated genes (e.g. TP63, KRT6A, S100A2), elevated LRP1B expression, increased TMB, and greater immune infiltration—features consistent with a more immunogenic and genomically unstable phenotype. In contrast, Cluster 2 was marked by higher expression of classical transcription factors (GATA6, HNF4A), consistent with a more progenitor-like and differentiated state. While most samples in our cohort exhibited lower transcriptomic gradient scores compared to conventional PDACs in the ICGC and TCGA datasets—consistent with an earlier-stage, classical phenotype typical of IPMN-derived pancreatic cancers—

the divergence between Clusters 1 and 2 highlights potential early commitment to distinct molecular programs.

These changes are now reflected in the revised Results and Discussion sections. We believe this integrated interpretation enhances the biological and translational relevance of the transcriptomic clustering analysis

7) The study mentions a "transcriptomic gradient score was assigned to each sample and compared to the ICGC PACA-AU20 and TCGA-PAAD cohorts" but lacks a clear explanation of how it is computed. This should be more explicitly described.

We thank the reviewer for this helpful comment. In the revised Supplementary Methods, we have clarified the computation of the transcriptomic gradient score. Specifically, single-sample gene set enrichment analysis (ssGSEA) was applied to each RNA-seq sample using curated gene signatures for the classical and squamous PDAC subtypes, as previously published by our group (Bailey et al. (Nature 2016) and Brunton et al. (Cell Reports 2020)). The final gradient score was calculated as the difference between the enrichment scores for the squamous and classical signatures. This approach allows for a continuous estimate of subtype identity across the epithelial transcriptomic spectrum. This information was added in the supplementary method section.

8) The study identifies novel CNAs and SVs, but their functional impact is not well explored. Have these structural variations been mapped onto known oncogenic pathways related to pancreatic cancer progression? The authors discuss LRP1B mutations and CNAs but does not provide sufficient functional interpretation. Could this gene's role in WNT signaling or immune modulation be further elaborated?

We thank the reviewer for this insightful comment. Our study was designed to address the underexplored contribution of structural variations (SVs) and copy number alterations (CNAs) in IPMN progression using multi-regional WGS—a level of resolution not available in prior IPMN studies based on WES or targeted sequencing.

We observed recurrent CDKN2A deletions, consistent with its known involvement in PDAC progression, and frequent RNF43 losses associated with high-grade lesions. While RNF43 mutations have been linked to dysregulated WNT signaling in IPMN, our identification of recurrent copy number loss extends these findings and supports the notion that biallelic inactivation may contribute to progression. U2AF1 deletions were also enriched in high-grade IPMNs, although their role in pancreatic tumorigenesis remains to be clarified.

Regarding LRP1B, this gene was both mutated and transcriptionally downregulated in Cluster 1—alongside higher tumor mutational burden, reduced WNT signaling activity, and increased CD8+ T cell infiltration. These findings are consistent with the immunogenic phenotype associated with LRP1B loss in other malignancies, and support a potential link between LRP1B inactivation, immune

evasion, and transcriptional reprogramming in a subset of IPMNs. However, our study did not experimentally validate the functional consequences of these alterations, and we agree this warrants further investigation.

Notably, although we did not identify specific SVs consistently enriched across high-grade IPMNs or PDACs, we observed a significantly higher SV burden in tumors with branching clonal evolution (as shown in Figure 2D). This suggests that SVs may play a key role in promoting subclonal diversification and genomic instability. This observation challenges prior models of IPMN progression largely based on SNV/indel data from WES, and underscores the added insight provided by WGS in revealing the structural complexity of these lesions.

We have therefore reinforced the potential role of SV in the discussion.

9) The authors reference previous work on IPMN evolution but should further contextualize their findings within existing models. How do the observed clonal architectures compare to those reported previously?

We thank the reviewer for this important comment. As outlined in prior responses and now expanded in the revised Discussion, we have acknowledged the contributions of previous studies and clarified how our work builds upon and extends this body of literature.

While earlier models based on targeted sequencing and WES demonstrated the polyclonal origin of IPMNs—primarily through the identification of distinct KRAS mutations within the same IPMN lesion—they largely inferred clonal heterogeneity from SNVs. These models proposed that early subclonal diversification drives progression through clonal selection during the transition to high-grade lesions, with substantial mutational heterogeneity observed only in LGD regions.

Our study builds upon and evolves these models by applying multi-regional WGS, enabling unbiased detection of structural variants SVs, copy number alterations, and non-coding mutations—classes of alterations not captured by WES. This approach allowed us not only to confirm polyclonality but also to demonstrate the complete absence of shared CNAs and SVs among genomically distinct clones within the same lesion, reinforcing a model of parallel evolution. Furthermore, we show that polyclonal architecture can persist into high-grade dysplasia and associated PDAC, suggesting that clonal divergence may extend further into tumorigenesis than previously appreciated.

We also note that prior studies employed Bayesian frameworks, such as the JAGS timing model (PMC7428044), to estimate the temporal dynamics of malignant progression. However, such models rely exclusively on SNVs and do not account for SVs or CNAs. Given the central role that SVs appear to play in IPMN evolution, as highlighted in our findings, we chose not to apply these timing models, as they may underestimate the complexity of clonal dynamics in the absence of whole-genome data.

10) The methods used to annotate phylogenetic trees with oncogenic alterations and mutational signatures are insufficiently described. The authors mention use of “in-house programs” to

integrate SNVs, indels, SVs, CNAs, and mutational signatures from various tools, but do not provide adequate detail on the algorithms, decision criteria, or confidence thresholds used to assign mutations to specific branches or subclones (no code, pipeline diagram, or methodological framework is described). This negatively impacts reproducibility and limits the reader's ability to assess the robustness of the phylogenetic annotations. For a high-impact publication in a journal like this, the authors should make the entire analysis code (for all analyses not just this one) available and provide a detailed methodological supplement explaining how different variant types were integrated and validated across the tree structure.

We applied both published and in-house-developed pipelines for phylogenetic annotation. To enhance clarity and reproducibility, we have added a flowchart (Figure S1A2) detailing each step in the Supplementary Methods section. Additionally, all analysis codes, including those for integrating SNVs, indels, SVs, CNAs, and mutational signatures, have been made available on <https://github.com/Wedge-lab/multidimDPCLust>, <https://gitfront.io/r/xtgitfhe2/eMoLG6vndJxF/IPMNpaperArchive/>.

11) The manuscript lacks details on how mutational signatures were assigned to tree branches. Were these based on clustered mutations, or aggregated per sample? Were exposures filtered for noise or assigned with confidence intervals?

We assigned mutational signatures to each branch in the phylogenetic tree using the following steps:

1. Mutation identification: we obtained the genomic coordinates of all somatic mutations from whole-genome sequencing (WGS) data.
2. Variant clustering: Using multidimDPCLust, we performed multidimensional clustering of variants, incorporating confidence intervals for assignment. This process generated mutational clusters along with their corresponding genomic coordinates. We included clusters containing at least 1% of the total mutations.
3. Phylogenetic reconstruction: we reconstructed the phylogenetic tree based on the outputs from multidimDPCLust, assigning mutations (along with their genomic coordinates) to specific branches.
4. Signature extraction: we identified single-nucleotide variant (SNV) signatures (SBS96) and insertion-deletion (Indel) signatures (ID83) using COSMIC methodologies. The genomic coordinates of mutations contributing to each mutational signature were extracted.
5. Assignment of signatures to tree branches: The probability of each mutational signature corresponding to a clonal cluster (tree branch) was derived by merging mutation coordinates from steps 3 and 4 for each multi-sample case. For each cluster, we used the probability of each mutation context being assigned to a signature and summed these probabilities across all mutations within the cluster.

All methods are detailed in the Methods section and Supplementary section - Annotation of the trees with mutations and signatures

(21-Assignment of Signatures to Clustered Branches:

<https://gitfront.io/r/xtgitfhe2/eMoLG6vndJxF/IPMNPaperArchive/>).

12) The description of subtype scoring is non-reproducible. It is not clear which gene sets were used or whether batch correction was applied before subtype classification.

We thank the reviewer for this observation. In the revised Supplementary Methods section, we have clarified the subtype scoring methodology and batch correction procedures. Specifically, transcriptomic gradient scores were calculated for each sample using single-sample gene set enrichment analysis (ssGSEA) based on previously defined classical and squamous gene signatures (Bailey et al., Nature 2016; Brunton et al., Cell Reports 2020). The final gradient score was computed as the difference between squamous and classical enrichment scores, with higher values indicating greater alignment with a squamous-like transcriptional program.

Batch correction was applied during normalization using the DESeq2 workflow, and potential confounders such as histological grade, epithelial subtype, and sequencing batch were considered in downstream analyses. This information has been added to the revised Supplementary Methods.

13) The methods describe numerous comparisons but there's no mention of multiple hypothesis correction, effect size thresholds, or the statistical models used. This weakens the credibility of reported p-values, especially for associations with cluster identity or progression status.

Thank you for highlighting the need for greater clarity in our statistical methods. In the revised manuscript, we now include details on the statistical models used for each analysis in the Supplementary method section: Data Processing and Statistical Analysis. Multiple hypothesis testing was controlled using the Benjamini–Hochberg procedure, with $FDR < 0.05$ considered statistically significant. Specifically, we applied Fisher–Pitman permutation tests to compare the number of SNVs, indels, and structural variants (SVs) across tumor types and genomic regions; Fisher's exact tests to compare the frequency of driver mutations (including SNVs, indels, SVs, and CNAs) and SV signatures across tumor types; and permutation-based Wilcoxon tests for pairwise comparisons of timing group proportions

14) The figures need significant improvement in terms of making them publication ready. They could benefit from additional annotations, particularly in phylogenetic trees to highlight key driver mutations at branch points. Some figures appear disconnected from the rest (Fig 2C) or hard to interpret (Fig 2B) due to lack of more information. For instance, the pie charts lack numeric annotations or mutation counts, making it hard to assess signature burden or dominance. The authors should add mutation counts or a bar/scale representing the number of SNVs/indels at

each node. Also for this same figure, it is unclear how signatures were assigned to branches. Are these inferred from the mutation cluster, or are they aggregated across constituent mutations?

Thank you for these helpful suggestions to improve the clarity and interpretability of the figures.

We have revised all figures to enhance their presentation quality and ensure they are publication ready. In the phylogenetic trees, we have added detailed annotations at key branch points to highlight the presence of known or putative driver mutations. This should help guide interpretation of evolutionary trajectories.

Figure 2B and 2C clarifications: To address concerns about Figure 2B, we have added numeric annotations and mutation counts within the pie charts to clearly reflect the signature burden and dominance.

For Figure 2C, we have clarified its relevance and integration with the rest of the analysis in the revised figure legend and main text, ensuring it is well-connected to the study's narrative.

Signature assignment clarification: Regarding the assignment of mutational signatures to branches, we have clarified this in both the figure legend and Methods section. Signatures were inferred based on aggregated mutational profiles from each cluster of mutations assigned to the corresponding branch.

We believe these updates address the reviewer's concerns and significantly improve the clarity and impact of the figures.

REVIEWER COMMENTS

Reviewer #1 (Remarks to the Author):

The authors have addressed my queries including the concerns about novelty and how this study differs from prior publications in this area.

We thank the reviewer for their positive feedback and are grateful that our revisions satisfactorily addressed their concerns.

Reviewer #2 (Remarks to the Author): cancer clonal evolution, structural variants and whole genome sequencing

The authors have addressed most of my concerns.

Consistent points:

- Some of the figures are still low resolution, part of the text is not visible (with different fonts and sizes), and overall they look "rushed" - which is odd and unusual to observe in a high-impact journal.

We thank the reviewer for this observation. All figure files have been carefully checked for consistency in format and layout to ensure clarity and reproducibility, and have now been re-exported and uploaded at high resolution (300 dpi). We trust that the revised figures now meet the standards of the journal.

- The pathway analysis is rather superficial - Fig S4/S5 - is there anything significantly different or conclusions to be drawn from this - instead of "notable differences"?

We thank the reviewer for this valuable observation. While some pathways appeared more enriched in invasive compared to non-invasive samples, only adipogenesis, androgen response, and heme metabolism were significantly enriched in non-invasive samples. We have revised the text accordingly and replaced Figure S5B with box plots highlighting these significantly enriched pathways.

- I could not access the code in the notebooks on the github (linked through gitfront), e.g. <https://github.com/xtgithubhe/IPMNPaperArchive/blob/main/ipmnPaperCode.ipynb>

Thank you for your comment regarding GitHub accessibility, and we apologize for any inconvenience caused by the link being provided without explanation. We have updated the GitFront link. All files in the repository, including ipmnPaperCode.ipynb, can be directly viewed online or cloned using the GitFront link provided in the manuscript:

<https://gitfront.io/r/xtgitfhe2/8f2rTdyZs1S9/IPMNPDAcPaperArchive/>

or

git clone <https://gitfront.io/r/xtgitfhe2/8f2rTdyZs1S9/IPMNPDAcPaperArchive/>

They contain exact duplicates of the previous private GitHub repository

(<https://github.com/xtgithubhe/IPMNpaperArchive/blob/main/ipmnPaperCode.ipynb>).

Once the manuscript is accepted, the GitHub xtgithubhe repository will be made public.

Reviewer #3 (Remarks to the Author): bioinformatics, cancer genomics, tumor clonal evolution, TME, TIME and transcriptomics

I appreciate the authors' revisions and thoughtful responses to the prior round of review. The addition of new analyses, expanded methodological descriptions, and improved code sharing through repositories represent meaningful progress.

However, after detailed examination of the revised manuscript, code repositories, supplementary materials, several important issues remain. These concerns relate to both scientific interpretation and reproducibility, and should be addressed to meet the expectations of a high-impact journal.

-Reproducibility and code sharing:

While the authors have shared parts of their pipeline through different repos, the full analytical workflow is not yet reproducible. Critical components remain unavailable:

a) Tree Annotation (SNVs, SVs, CNVs, Signatures): No public scripts are provided for mapping mutations and mutational signatures onto tree branches. The process is described narratively but is not executable by readers.

Thank you for this useful comment. We have now updated the methods and provided the method and executable code for tree annotation (SNVs, SVs, CNVs, and mutational signatures). To ensure reproducibility, we have shared both the running code and relevant example datasets in the same repositories:

<https://gitfront.io/r/xtgitfhe2/8f2rTdyZs1S9/IPMNPDAcPaperArchive/>

As outlined step by step:

1. Subclonal clustering: We applied *multidimDPClust* to the IPMN–PDAC WGS data, which is specifically designed for multi-region WGS samples (<https://github.com/Wedge-lab/multidimDPClust>).

2. Phylogenetic tree inference: Based on the identified SNV/indel clusters, we determined the most likely tree branches using the “pigeon-hole” principle (PHP), the “sum rule,” and the “crossing rule,” as detailed in references ^{10–12}. We have highlighted this procedure in the **Supplementary Methods** (section: **Mutation clustering and phylogenetic tree reconstruction**).

3. Tree annotation of SV: SV clusters were identified by matching their genomic coordinates with those of SNV/indel clusters from the same sample. SV drivers were then assigned to the corresponding branches (see revised section **Tree annotation of SV and CNV** of the Supplementary Methods and analysis code at:

24-Assignment of SV (gene-driver) to Clustered Branches,

<https://gitfront.io/r/xtgitfhe2/8f2rTdyZs1S9/IPMNPDACpaperArchive/>).

4. Tree annotation of CNV: Similarly, CNV clusters were assigned to tree branches by genomic coordinate matching to SNV/indel clusters (see revised section **Tree annotation of SV and CNV** of the Supplementary Methods and analysis code at:

25-Assignment of CNV (gene-driver) to Clustered Branches,

<https://gitfront.io/r/xtgitfhe2/8f2rTdyZs1S9/IPMNPDACpaperArchive/>).

5. Mutational signature annotation: In addition to the description in the **Annotation of the trees with mutations and signatures** section of the Supplementary Methods, we have now updated and shared the analysis code for assigning signatures to tree branches:

(21-Assignment of Signatures to Clustered Branches:

<https://gitfront.io/r/xtgitfhe2/8f2rTdyZs1S9/IPMNPDACpaperArchive/>).

Together, these updates provide readers with a full method to reproduce the tree annotation steps.

b) Transcriptomic and Immune Integration: The analyses of transcriptomic clustering, subtype assignment, and immune deconvolution are described but not accompanied by code.

We thank the reviewer for this important comment. The full script for the transcriptomic and immune analyses has now been uploaded at

<https://gitfront.io/r/xtgitfhe2/8f2rTdyZs1S9/IPMNPDACpaperArchive/> and includes data input,

clustering, ComBat-Seq correction, subtype assignment, cancer hallmark analysis, and immune deconvolution (EPIC).

c) Shared code through different repos reference local paths with no standardized data input structure or templates provided for external reproducibility.

Overall, while parts of the pipeline are transparent, the complete analytical workflow is not yet reproducible. For a study of this scope, particularly one centered on complex evolutionary modeling, end-to-end reproducibility is essential and everything should be unified in a single repo for users instead of pointing readers to different repos for different parts.

We are grateful to the reviewer for highlighting the importance of reproducibility. We have unified all analysis code into a single GitHub repository:

(<https://gitfront.io/r/xtgitfhe2/8f2rTdyZs1S9/IPMNPDAcPaperArchive/>)

This repository contains a standardized input directory along with relevant datasets required to reproduce the results. It also includes wrapper scripts covering the entire method from data processing and statistical analysis to evolutionary modeling, mutation assignment to phylogenetic branches, and figure generation.

Furthermore, we have updated the integration analysis of genomic, transcriptomic, and immune data within the same repository. This includes dataset preparation, clustering, ComBat-Seq-corrected expression data, subtype assignment, cancer hallmark annotation, and immune deconvolution (EPIC).

We believe these updates address the reviewer's concerns and ensure end-to-end reproducibility of our study, while also providing a more integrated view of genomic evolution, transcriptomic subtypes, and the immune microenvironment.

-Integration of Genomic, Transcriptomic, and Immune Data:

Although new immune analyses were added, the study still lacks mechanistic integration between genomic evolution, transcriptomic subtypes, and immune microenvironment features. The relationships remain correlative. For example, there is no test of whether specific mutational patterns predict transcriptional or immune shifts.

We thank the reviewer for this constructive comment. In the revised version, we strengthened the integration between genomic evolution and immune contexture by explicitly comparing immune cell composition across clonal trajectories. We found that CD8⁺ T-cell infiltration and ESTIMATE Immune Scores were significantly higher in tumors evolving from a single MRCA

compared to those with multiple independent clones, suggesting that clonal architecture modulates immune surveillance.

In the Discussion, we now interpret this finding by proposing that single-MRCA tumors retain uniform clonal alterations that may favor immune recognition, whereas multiple independent clones generate distinct, lineage-specific alterations that fragment and dilute antigenic visibility, thereby contributing to immune evasion. Although we did not directly assess neoantigen presentation, this interpretation is consistent with prior studies showing that tumors enriched for clonal rather than heterogeneous neoantigens are associated with stronger immune responses. We acknowledge that this remains a hypothesis, which we plan to address directly in future studies through integrated neoantigen prediction and functional validation.

-Figures are partially improved but some, especially Fig 2, lacks critical quantitative features and full integration of genomic events:

a) Fig 2a: No branch length scaling is provided; trees are topological, not quantitative.

b) Fig2b: The SV counts across branching architectures are presented, but no mapping of specific SVs onto the phylogenetic trees is performed.

c) Fig2c: the circos plots are of extremely low resolution, and labels are unreadable.

Furthermore, SVs are displayed per sample but are not linked back to the clonal architecture in Fig 2D.

d) Fig2d: Branch lengths are absent. For a study claiming to map clonal trajectories and mutational processes over time, branch lengths are essential. Also, SVs and CNVs are not mapped onto the trees, despite their importance to the clonal evolution narrative.

Thank you for your valuable comments. To improve the figures, we have made both scientific and technical revisions by adding quantitative features and optimizing visibility.

a) Figure 2a: We have now provided branch length scaling. Specifically, the legend at the bottom indicates the length scale based on the total number of SNVs and indels (with units of ~150, 200, and 1000 mutations). Additionally, for clarity, we have labeled the number of mutations on the longest branch in each tree, which will help readers estimate the relative lengths of all other branches.

b) Thank you for this suggestion. We have now mapped cancer driver SVs onto the corresponding branches of the phylogenetic trees in Figures 2A and 2D, thereby directly linking the SVs to the clonal architecture.

c) We have improved the quality of the circos plots in Figure 2C by substantially increasing their resolution and adjusting the labeling for better readability. Furthermore, to integrate these data

with the clonal evolution framework, we have mapped the cancer driver SVs, including those displayed in the circos plots, onto the phylogenetic trees shown in Figure 2D.

d) We appreciate your comment and fully agree on the importance of branch lengths, CNVs, and SVs in illustrating clonal evolution. Fig. 2d is intended as an unscaled schematic focusing on mutational signature profiles within the clonal trajectories and is largely derived from Fig. 2b. In Fig. 2b, we have already included scaled branch lengths and mapped CNVs and SVs. For improved readability in Fig. 2d, we have now labelled the key driver events related to CNVs and SVs on each branch and indicated in the legend that branch lengths can be referenced from Fig. 2b. We hope this modification addresses your concern.

REVIEWER COMMENTS

Reviewer 3:

Provide a README.md or tutorial describing end-to-end use of the shared repository, including required data formats, example commands, and environment setup.

We thank the reviewer for this helpful suggestion. We have added a comprehensive README.md file to the shared repository. This document includes detailed instructions for environment setup, input data formats, and example commands to enable full end-to-end reproducibility of the analyses presented in the manuscript.

Consider a supplementary figure or model linking mutational features to immune phenotypes in a mechanistic way.

We appreciate this suggestion. Figure 2A now displays all genomic alterations—including SNVs/indels and structural variants—mapped onto the phylogenetic branches for each case, and the case order has been aligned with Figure 3A, which defines the transcriptomic clusters. This alignment ensures consistency between figures and text, and allows readers to visually integrate clonal architecture, genomic alterations, and the corresponding transcriptomic and immune phenotypes within a coherent narrative framework.